# Youth, personality and collective victimhood distinguish support for radical climate action

Matthew J. Hornsey [1] ✉, Samuel Pearson [1], Susilo Wibisono[1,2], Emma F. Thomas [3], Lucy H. Bird[3,4], Jarren L. Nylund [1], Christian Bretter [1], Janquel D. Acevedo [1], Kelly S. Fielding[1], Catherine E. Amiot[5], Fathali M. Moghaddam [6] & Winnifred R. Louis [1]

Despite the fact that law-breaking or violent climate action tactics receive enormous media coverage, the psychological predictors of intentions to engage in these tactics remain poorly understood. This study examined demographic and psychological factors theoretically associated with conventional and radical climate intentions among 1427 self-identified supporters of climate action, tracked in three waves over 12 months. Conventional activism intentions were predicted by established models emphasising the role of moral conviction, anger, group identification, and group efficacy in shaping action. However, in the case of radical climate action, these variables were either weak predictors or non-significant predictors. Contrary to the notion that radical climate actors are driven by outgroup antipathy and ideological intensity, radical action intentions were positively associated with warmth and empathy toward climate change opponents, unrelated to political ideology, and negatively related to belief in climate change. Radical action intentions were also predicted by youth, personality, and—most strongly—the perception that people who support action on climate change have suffered more than opponents (collective victimhood). These findings suggest that theories require updating to account for the unique motivations associated with support for radical tactics in the climate change context. Findings have implications for activists and researchers seeking to understand the evolving landscape of climate protest and public support for disruptive activism.

Despite half a century of intense scientific and political communication about the threat of climate change, little progress has been made in reducing carbon emissions. Since 2007, humans have emitted over a quarter of all the greenhouse gases ever emitted by our species, and in 2024, global $CO_2$ emissions hit a record high. In response, many people have engaged in climate collective action, defined here as efforts by groups or communities to address climate change through advocacy, policy change or direct action. Engaging in *collective* action, as opposed to individual action, makes sense because collective action has multiplier effects on policy and practice that are commonly presumed to surpass the impact of disjointed individual efforts[1–3]. Indeed, examples of collective action are often celebrated among those concerned about climate change as emblems of hope, pride and positivity[4].

A long tradition of research has distinguished between forms of collective action that are *conventional*—that is, lawful, socially sanctioned and aligned with prevailing norms—and *radical* actions that are illegal, disruptive, violent or otherwise outside institutional rules[5,6]. This distinction closely parallels terminologies reflecting differences between *normative versus non-normative*[7], *activism versus radicalism*[8], *legal versus illegal*[9] and *hostile versus benevolent* action[10]. Importantly, prior work shows that different forms of action are shaped by different psychological pathways.

For example, Selvanathan and Leidner[11] demonstrate that normative and non-normative actions emerge from distinct constellations of identity and justice concerns: ingroup attachment predicts normative action via restorative justice motives, whereas ingroup glorification predicts non-normative action via retributive motives. The distinction has also been explored in the environmental domain: Bashir et al. find that the public differentiates between 'typical' and 'atypical' environmentalists, and that atypical (i.e. more extreme or radical) actors elicit social distancing and

[1]University of Queensland, Brisbane, QLD, Australia. [2]Murdoch University, Perth, WA, Australia. [3]Flinders University, Adelaide, SA, Australia. [4]James Cook University, Townsville, QLD, Australia. [5]Université du Québec à Montréal, Montreal, QC, Canada. [6]Georgetown University, Washington, DC, USA. ✉e-mail: m.hornsey@uq.edu.au

reduce willingness to engage in or support environmental action[12]. Although various terminologies have been used across domains, they converge on a shared insight: people appear to draw categorical distinctions between conventional actions that work *within* established systems and radical actions that seek to disrupt or challenge them. What remains unclear, particularly in the climate context, is whether these distinctions reflect incrementally stronger engagement along a single continuum, or whether they represent qualitatively distinct forms of action associated with different motives. Our study directly addresses this gap.

Exemplifying the radical action approach, prominent climate groups such as Extinction Rebellion, Last Generation, and Just Stop Oil have engaged in acts of civil disobedience and public disruption[13]. Tactics of protesters include affixing themselves to infrastructure, vandalising property with food and paint, sabotaging oil and gas pipelines and obstructing key transport routes, including bridges, highways and roads. These actions frequently cause public disruption and attract enormous media attention[14]. Support for such tactics is relatively high among academic commentators, to the point that several high-profile pieces have called for scientists to go beyond traditional outreach activities to engage in civil disobedience themselves[15,16]. Calls for civil disobedience typically stop short of advocating violence[17], although invocations of violence are occasionally circulated in the literature[18] and 'green terrorism' is sometimes nominated as a live threat by security agencies[19].

Despite attracting a disproportionately large amount of media attention and having an outsized impact on both public and academic discourse, we currently have a limited understanding of the psychological predictors of radical climate action intentions. As elaborated below, most psychological theories of collective action focus on conventional action, and there is limited evidence that speaks specifically to radical action. Theories of radicalisation that do exist are typically generated to explain radical action in unambiguously intergroup contexts (e.g. civil rights, labour strikes, geopolitical contests over occupied lands). It is an empirical question whether these theories can be readily generalised to the climate change context.

## Theory and research on conventional collective action

The most widely applied psychological model to understand why people engage in collective action is the social identity model of collective action (SIMCA)[20]. SIMCA has received various elaborations and extensions over the years—for example, the encapsulated model of social identity in collective action (EMSICA)[21] and the social identity model of pro-environmental action (SIMPEA)[22]. Recent theoretical developments—for example, the disidentification, innovation, moralisation and energization model (DIME)[6] and the model of belonging, individual differences, life experience and interaction sustaining engagement (MOBILISE)[23]—build on insights from SIMCA while also emphasising dynamic and processual aspects of mobilisation such as tactical innovation, resourcing, and feedback loops.

However, these perspectives share four core factors that are presumed to drive intentions to engage in collective action. *Perceived injustice* refers to the belief that one's group is unfairly treated or oppressed, and affective responses to this belief such as anger[7,24,25]. *Moral conviction* is the belief that the group's cause is not just a preference but a matter of right and wrong[10,26–28]. This moral clarity strengthens the drive to engage in collective action, even when the risks are high. *Social identification* refers to the extent to which individuals psychologically feel connected with a group engaged in collective action. The stronger their identification, the more likely people are to feel group cohesion and act in support of the group's goals[22,29]. *Collective efficacy* refers to the belief that the group people identify with will be effective in achieving its goals. If people think the movement can make a difference, they are more likely to engage in activism[24,30]. The empirical case for the role of these four factors in predicting conventional collective action is strong, having been validated by multiple meta-analyses and systematic reviews[20,31,32].

## Theory and research on radical collective action

The models described above were primarily designed to explain conventional collective action, and the empirical work that tests them overwhelmingly focuses on conventional forms of protest. Implicit in this work, however, is that more radical action can arise when the standard predictors are experienced at very high levels. One example is when perceived injustice is framed as severe and urgent, justifying extreme measures. Another example is when identification with the movement is reinforced by radicalising small group dynamics and/or intensifies to the point that individual selves psychologically fuse with the collective[33,34].

Subsequent theorists have proposed three factors that might lead to radical collective action over and above the themes described in SIMCA. First, whereas conventional collective action is associated with high levels of collective efficacy, radical collective action is theorised to be associated with low levels of collective efficacy[35,36]. When people feel so marginalised that they perceive they have no personal or collective power within the current system, they become more willing to engage in radical collective action (this is sometimes described as a 'nothing-to-lose' mentality)[37]. Second, radical action is theorised to be associated with high levels of outgroup antipathy. Whereas anger is associated with conventional collective action, radical action is more likely to be associated with feelings of contempt (viewing authorities as corrupt and beyond redemption) and hatred (deep dehumanisation of the perceived oppressor)[7,38]. Third, radical action is theorised to be associated with perceptions of collective victimhood. Movements that see themselves as both virtuous and victimised may justify aggression toward perceived enemies[39]. When combined with a strong sense of humiliation or historical injustice, this can escalate into violent protests, destruction of property, or attacks on outgroups[40]. In short, people are presumed to turn to radical action when they feel the political system is rigged against them, making traditional activism futile, and when they feel deep outgroup antipathy.

## Radical climate action

Currently, both the theoretical and empirical basis for understanding radical climate action is somewhat fragile. In terms of empirical work, three studies focused on climate activists: one survey on 203 current or potential Extinction Rebellion activists in the UK[41], one on 223 Germans recruited via snowball sampling of university mailing lists and activist social media platforms[36], and a snowball sample of 253 German psychology students recruited through social media and networks of the researchers[42]. These studies underscored the role of politicised identity, perceived injustice, and collective efficacy as motivating forces for radical climate action. We are aware of only four empirical studies that have measured the psychological correlates of radical climate action intentions in the general population (as is the focus of the current study). One study focused on short- versus long-term orientations[43], one focused on eco-fascism[44] and another focused on Dark Triad personality variables[45]. More relevant to our theoretical approach, a survey study of psychology undergraduates revealed that radical collective action intentions about climate change correlated with illegitimacy, political despair, and anger, although none of these variables correlated with self-reported radical action behaviour[46].

In terms of theorising, it is notable that most psychological accounts of radical action are grounded in overtly intergroup contexts[7,47]. Concepts such as outgroup antipathy presume there is a clearly defined outgroup, and 'nothing-to-lose' accounts presume that there is a clear, consensual sense of the nature of the *status quo* and who is in the ascendancy. Although many social movements involve multiple and intersecting fronts, debates around climate change are arguably marked by particularly diffuse and shifting intergroup boundaries. Rather than coalescing around a single, historically entrenched identity cleavage, climate action draws upon—and cuts across—political, national, generational, and sectoral fault lines. This fluidity means that potential participants are not always sure who they are standing with or against, which may dampen the potency of traditional group-based drivers of action (e.g. strong identification with an aggrieved ingroup versus antagonism toward a clear outgroup). Thus, it remains an open and important question whether motivational pathways established in more identity-centric movements apply fully to the climate context.

As such, existing research and theorising are possibly not fit-for-purpose to explain radical action in the climate change sphere[48]. For example, in the struggle over climate change, the intergroup context is relatively amorphous. For supporters of climate action, 'ingroups' represent anybody who shares that commitment to action, a broad constituency that incorporates a soft majority of citizens who support action but do not self-identify as activists; progressive activists who see action on climate change as an opportunity to dismantle capitalism and reduce historical power differentials[49]; and eco-fascists who might think the opposite[44]. Unlike traditional intergroup conflicts, climate activists advocate not just for a specific social group but for humanity as a whole and the natural world—a scope that transcends conventional intergroup frameworks[50].

For climate activists, the outgroup might be equally amorphous, comprising not only individuals who resist climate-positive change, but also various institutional powers (e.g. fossil fuel corporations and obstructionist governments) perceived as deliberately orchestrating delays in decarbonisation to protect their economic and political interests[51,52]. As such, the battle over climate mitigation is complicated by multiple intergroup boundaries and fought on multiple fronts[53]. Theories of radical action that presume outgroup antipathy might not be well-suited to explain radical action where the intergroup boundaries are so complex and intersecting. Similarly, law-breaking that has been studied in other environmental contexts, such as animal liberation or clear-cutting[35,54], typically involves localised intergroup dynamics with defined opponents, whereas climate-focused radical actions address a global and multi-factor system without a single adversary.

Furthermore, theories that focus on the illegitimacy and stability of the status quo might not be well-suited to the climate change context, where there is not always a clear consensus about who is in the ascendancy. Many people point to the dominance of fossil fuel companies and the continued rise in greenhouse gas emissions as evidence that opposition to change remains steadfast or growing stronger[55]. However, some opponents of climate action make the reverse case: that governments have been captured by Big Green vested interests that have pushed through large-scale regulations and investments that are unnecessary and in contradiction to 'evidence'[56]. Presumably, people in the middle ground vary along this spectrum. The fluid and non-consensual perceptions of the intergroup context make climate change a qualitatively distinct case than some other intergroup contexts (e.g. when people resist occupation from a foreign power, or activists seek to overturn legislation).

## The current study

In response to these theoretical and empirical uncertainties, we measured a theory-driven set of demographic and psychological predictors of intentions to engage in different forms of climate action. Our sample comprised 1427 community members who self-identified as supporting action on climate change. This study was conducted in Australia—a context that has historically experienced high levels of public concern about climate change alongside protracted political polarisation and policy volatility[57–59]. Such dynamics may shape both the salience of climate action and the psychological drivers of activism intentions, and we return to these contextual factors in interpreting the findings.

Unlike some previous studies, we aimed for a sample of the general population rather than a sample of activists or members of activist groups. For ethical reasons, we did not ask people to report whether they had broken the law or engaged in violence, but rather whether they intended to engage in these acts or intended to support organisations that engaged in these acts (including financially)[8].

We acknowledge that participation in collective action and intentions to engage in collective action are conceptually distinct (as participation typically entails higher levels of personal cost and involvement). Nevertheless, we follow the majority of the literature on this topic[20,31] in measuring intentions as a reliable (but imperfect) proxy for behaviour. Given that public backing is an important precursor to action and

remains comparatively under-theorised, this reflects a deliberate theoretical extrapolation rather than a conflation of constructs.

Predictors included in the model, along with the theoretical bases for their inclusion in the analysis, are summarised in Table 1. All items and scale reliabilities are detailed in Supplementary Table S1. Age, sex and education were included as control variables due to their documented systematic associations with the focal predictors in prior research. Predictors represented five broad themes: (1) climate-related conviction (i.e. belief in anthropogenic climate change and moralisation of those beliefs), (2) emotions about climate change (i.e. anger), (3) beliefs about the ingroup (i.e. identification and perceptions of collective efficacy), (4) perceptions of the intergroup context (i.e. perceptions of state capture, collective victimhood, norms of outgroup hostility, outgroup empathy, and outgroup favourability), and (5) individual differences (i.e. political ideology and personality). Our personality constructs were short-form measures of the Big 5[60], inclusion of which was motivated by research showing that climate mitigation intentions and pro-environmental behaviours are greater among those higher in openness to experience, agreeableness and conscientiousness[61]. However, support for radical collective action specifically has been shown to be associated with lower conscientiousness[45] and higher neuroticism[44], suggesting a complicated and still-unresolved role for the Big 5 personality traits in understanding collective action intentions. Political ideology was measured due to the long-held political polarisation in Australia on climate change issues[59,62,63], and international research indicating people on the left end of the political spectrum are generally more willing to engage in pro-climate collective action[64].

## Methods

This study received ethical approval from the Institutional Review Board of The University of Queensland (2023/HE000466) and was conducted in line with the ethical guidelines of the National Health and Medical Research Council. All participants provided informed consent. We paid ORU AUD $7 for each participant for each wave, although the exact amount respondents received is subject to ORU policy. This study was not pre-registered.

### Sampling and exclusions

Participants were sampled in June 2023 and then recontacted twice to complete follow-up surveys (Wave 2 in December 2023; Wave 3 in June 2024). Participants were recruited using the Online Research Unit (ORU), an Australian data collection company that builds online panels using both offline recruitment (telephone, print and postal) and online recruitment. Participants were told that 'the purpose of this study is to understand the psychological processes affecting decision making in the context of debates on climate change mitigation and policies'.

The follow-up surveys comprised the measures we used in the Wave 1 survey, with the exception of demographics and personality measures, which were presumed to be relatively stable. Matching across samples was facilitated by an ORU personal identifier and a unique identifier that the participants generated. Responses were screened out if they were suspected of being bots on the basis of failing Qualtrics' Captcha screen (RecaptchaScore < 0.50) and a honeypot item ('Are you human?' Yes/No). We also screened out those who completed the survey in less than 5 min, respondents who completed less than 50% of the survey, respondents who reported they were under 18 years of age, respondents who did not consent to continue the study after the information sheet, and respondents who, after completing the survey, responded affirmatively to a question about whether they would like their data be withdrawn from analysis. Finally, we excluded respondents who failed either of the two attention checks ('To show that you are paying attention, please select 2'). The initial sample was representative in terms of gender (1024 female, 1002 male, 4 non-binary or 'other', and 2 respondents who preferred not to indicate their gender). The average age of our sample ($M = 52.84$, $SD = 16.95$) was slightly higher than the Australian population (approximately 46 years when excluding those under 18). The average score on political ideology ($M = 3.90$) approximated the midpoint of the 1 (left-wing) to 7 (right-wing) scale.

**Table 1 | Theoretical prediction of relevant constructs and collective action**

| Construct | Theoretically derived prediction |
|---|---|
| *Identification*—the extent to which individuals feel psychologically connected with a group engaged in collective action | The stronger their social identification, the more likely people are to act in support of the group's collective action goals (SIMCA, EMSICA, SIMPEA and MOBILISE). |
| *Collective efficacy*—the belief that collective action will be effective in achieving its goals | If individuals think their group can enact change, they are more likely to support activism (SIMCA, EMSICA, SIMPEA and MOBILISE). However, when people feel low efficacy, especially through failure, they become more willing to support radical action (DIME). |
| *Moral conviction*—the belief that the group's cause is not just a preference but a matter of right and wrong | Moralisation of one's beliefs about climate change predicts increased willingness to engage in collective action (SIMCA, EMSICA, SIMPEA and MOBILISE). Heightened moral conviction, especially under failure, can lead to more willingness to engage in radical collective action (DIME). |
| *Anger*—operationalised here in terms of the anger people feel when thinking about the current debates on climate change | Anger, mainly due to perceived injustice, predicts engagement in collective action (SIMCA, EMSICA, SIMPEA and MOBILISE). |
| *Outgroup antipathy*—operationalised here in terms of (low) empathy and (low) favourability toward opponents of climate change | Antipathy toward the outgroup is theorised to be associated with radical action (Becker and Tausch)[7]. |
| *Norm of hostility*—the extent to which other ingroup members are perceived to be hostile to the outgroup | Ingroup norms shape individual action when it comes to responding to climate change (SIMPEA) |
| *Belief in climate change*—the belief that climate change is real and caused by humans | Belief in anthropogenic climate change is a pre-requisite for engaging in collective efforts to mitigate emissions. |
| *Political ideology*—operationalised here in terms of the extent to where people self-identify, politically, on the spectrum of left-right | Left-wing political orientation is associated with more pro-climate attitudes and collective action intentions (AICAM and MOBILISE). |
| *Collective victimhood*—the tendency of groups to claim they have suffered more than an opposing group | Collective victimhood is analogous to perceived injustice and predicts collective action (SIMCA, EMSICA, SIMPEA and MOBILISE). However, movements that see themselves as virtuous and victimised may justify aggression toward perceived enemies and radical action. |
| *State capture*—the perception that powerful actors manipulate government and institutions to serve their interests over the public good | People turn to radical action when they feel the political system is rigged against them (Becker and Tausch)[7]. |
| *Big 5 personality*—the set of stable traits and behaviours that shape how individuals interact with their social environment | Personalities directly and indirectly affect engagement in collective action through group consciousness (MOBILISE; Zacher)[45]. |
| *Demographics*—measured here as age, gender, and education | Demographic characteristics directly and indirectly affect collective action through group consciousness. For example, age and gender have been relevant in the past to predict collective action (MOBILISE). |

Predictions were not pre-registered.

After entering their demographics, participants were asked, 'Overall, thinking about your own position on climate change, which statement would you say better describes yourself?' Participants were given three options. Our analyses are based on the 1427 valid respondents who chose the option 'I support taking action to address climate change'. Those who chose the option 'I do not have an opinion on this issue/I have mixed feelings / I stay out of it' ($n = 506$) or 'I oppose taking action to address climate change' ($n = 202$) were not analysed for this paper. The final sample comprised 739 women and 683 men; 5 respondents who reported being non-binary or preferred not to reveal their gender were treated as missing data in analyses featuring gender. The average age of the sample of supporters was 51.45 years ($SD = 16.96$), and the mean political ideology was 3.55.

Of the 1427 supporters of climate action analysed in the first wave, 599 (42.0%) completed the Wave 3 survey. Attrition analyses, reported in Supplementary Tables S2a–c, showed that completers were older and (when comparing Waves 1 and 2) had stronger beliefs in climate change. The survey was highly powered: G*power analysis identified that, for 20 predictors in a standard regression, $N = 324$ would be sufficient to be 95% confident of detecting a small effect ($f^2 = 0.1$).

**Measures**

The full set of items and information about scale reliability are summarised in Supplementary Table S1. The sets of measures were completed in the following order: demographics; Big 5 personality; outgroup favourability; belief in climate change; moral conviction; conventional collective action; radical collective action; collective victimhood; anger; empathy toward the outgroup; collective efficacy; norms of hostility; state capture; and identification. Within each of these blocks of measures, the order of the specific items was randomised. Correlations among measures are reported in Supplementary Figs. S1 and S2. Distribution of scores and central tendencies is graphically displayed in Supplementary Fig. S3.

As shown in Supplementary Table S3, the exploratory factor analysis produced a clear two-factor solution corresponding to conventional and radical collective action intentions. All items loaded most strongly on their theorised factor, with two items showing secondary cross-loadings of interpretable magnitude. The 'participate in a sit-in' item loaded primarily on the conventional factor ($\lambda = 0.58$) but also showed a modest secondary loading on the radical factor ($\lambda = 0.40$), consistent with the ambiguous status of sit-ins in the literature, where they are variously classified as conventional, disruptive, or borderline non-normative depending on context[17,65,66]. Likewise, the item 'donate to an organisation that sometimes breaks the law' loaded primarily on the radical factor ($\lambda = 0.51$) but showed a weaker secondary loading on the conventional factor ($\lambda = 0.43$), reflecting its combination of a conventional action form (donation) with radical organisational behaviour. Because both items loaded most strongly on their intended factor, contributed to high internal reliability within each scale, and mapped onto theoretically coherent distinctions, they were retained in accordance with recommendations for construct-valid scales with conceptually interpretable cross-loadings. Means and standard deviations for each of the collective action items are summarised in Supplementary Table S4.

**Statistical analyses**

We first used multilevel models to examine determinants of conventional and radical collective action over timepoints using the 'lme4' package for R software. We modelled collective action across three measurement waves, with person-level predictors including demographic factors, personality traits, and psychological constructs related to collective engagement. Specifically, we fit a random intercept model where all predictors were specified at the between-person level, with time-varying psychological constructs aggregated to person means. This approach allowed us to examine how stable individual differences predict collective action tendencies while accounting for the non-independence of repeated

measurements. Standardised coefficients were estimated to facilitate comparison of the relative importance of different predictors.

To examine longitudinal relationships, we used dynamic structural equation modeling (DSEM) using Mplus version 8.11. This approach extends the random-intercept cross-lagged panel model by distinguishing between stable, trait-like differences between people and dynamic, within-person processes across time. In a three-wave design such as ours, DSEM estimates autoregressive effects (how a variable predicts itself over time) and cross-lagged effects (how one variable predicts change in another at the next wave), while simultaneously modelling individual differences in these dynamics. This provides a clearer picture of both the stable and time-sensitive processes that shape collective action intentions.

Given the longitudinal focus of the DSEMs, we retained only those participants for whom we had complete responses at all three waves. In addition, several of our predictors (e.g. personality, political ideology) were treated as stable dispositions and measured only at baseline, meaning they could not be included in longitudinal DSEMs. Finally, attempts to include all remaining predictors in a single model led to convergence problems. We therefore estimated separate bivariate DSEMs for each longitudinal predictor that showed significant effects in the multilevel models, allowing us to test their dynamic associations with collective action intentions over time. All analyses used two-tailed tests of significance.

## Assumptions of statistical tests

All analyses were conducted in line with recommended practice for psychological survey data. For the multilevel models predicting conventional and radical collective action intentions, visual inspection of residuals indicated that assumptions of linearity and homoscedasticity were reasonably met. Distributions for most predictors and for conventional collective action intentions approximated normality. However, as noted in the 'Results', radical collective action intentions were highly skewed with a marked floor effect. Although multilevel models are generally robust to non-normality in residuals, this distributional constraint limits the precision of linear estimates for this outcome. To address this, we supplemented the regression models with non-parametric comparisons (independent-groups t-tests using a theoretically informed cut-point) to ensure that key patterns were not artefacts of model assumptions. For the DSEM analyses, normality of the latent within-person components is assumed, but the extreme floor effects on radical intentions likely contributed to the absence of detectable longitudinal effects. These limitations are acknowledged in the 'Discussion'.

## Results

Overall, conventional collective action intentions were below the scale midpoint ($M = 3.16$, $SD = 1.28$) but there was substantial variability in views, with approximately a quarter of respondents (25.3%) scoring above the scale midpoint. In contrast, there was low reported intention to engage in radical collective action. The mean was near-floor ($M = 1.62$, $SD = 0.98$) and only 3.9% (56 of 1,427 self-identified supporters) scored above the scale midpoint. Indeed, nearly half the sample (45.9%) recorded a score of 1 ('strongly disagree') on all six items of the radical collective action scale.

The core of the analyses involved mapping associations between the two dimensions of collective action and our 19 predictors. We measured collective action intentions at three time-points: first in June 2023 (in the same survey from which the predictors were measured), and then twice more at 6-monthly intervals. Examining the extent to which the predictors explained variance in collective action intentions over time extends the cross-sectional analyses in two valuable ways. First, it highlights the resilience in factors that predict collective action intentions over time. Second, it reduces the contaminating influence of common method variance and response biases (e.g. respondents circling higher numbers all the way through the survey, inflating correlations among positively worded items). Third, it provides clues as to temporal precedence of variables.

Figure 1 summarises the results of multilevel models examining the value of our variables in predicting conventional collective action intentions over the three waves. Specific statistics are detailed in Tables 2 and 3.

Significant predictors of conventional collective action—in descending order of effect size as indexed by the absolute values of the standardised regression values ($\beta$)—were collective victimhood, identification, moral conviction, collective efficacy, youth, (low) conscientiousness, outgroup empathy, political conservatism (more left-wing participants reported stronger intentions), anger, education, openness to experience and extraversion. In short, the pattern of results supported most of our theoretically derived expectations (Table 1).

Figure 2 summarises the results of the same analyses repeated for radical collective action. Significant predictors (in descending order of standardised coefficient) were: collective victimhood, (lower) belief in climate change, youth, (low) conscientiousness, outgroup favourability, gender, identification, outgroup empathy, (low) agreeableness, and moral conviction.

Although some of these predictors were anticipated by Table 1, there were several noteworthy exceptions. For example, radical action intentions were not significantly associated with political ideology or collective efficacy. Furthermore, there was limited evidence that radical action intentions were associated with strong beliefs about climate change: indeed, radical action intentions were associated with *lower* belief in anthropogenic climate change. Finally, in contrast with assumptions about radical action being associated with outgroup contempt, radical action intentions were associated with greater favourability and empathy toward opponents.

Interpreting the regression on radical collective action intentions is compromised by the presence of low scores overall, creating a positive skew that was unable to be corrected by log, square root or inverse transformation. To account for this, we ran a series of independent-groups t-tests comparing the 56 participants who scored above the midpoint of the June 2023 radical action scale with the participants who scored on or below the midpoint. As can be seen in Table 4, the profile of radical actors that emerged in regressions was largely replicated in the t-tests: radical actors (compared to non-radical actors) were younger; lower in agreeableness and conscientiousness; higher in extraversion; more likely to be male; lower in belief in climate change; relatively positive in their views about the outgroup; but were more likely to perceive a norm of hostility toward outgroup members. As in the regressions, the strongest effect was that radical actors were more likely to perceive collective victimhood than non-radical actors.

## Examining longitudinal associations

To analyse change over time, we used DSEM. This approach not only examines whether predictors and outcomes are correlated but also tests whether fluctuations in one variable forecast subsequent changes in another. By separating stable between-person differences from dynamic within-person processes, DSEM provides a sharper lens on why collective action intentions rise or fall across time. In practice, this meant we ran simpler two-variable (bivariate) models rather than one large, all-in model (see Methods). We focused only the variables that emerged as significant in Figs. 1 and 2 and were measured across all three waves.

While our primary DSEM models tested the extent to which psychological variables predicted change in collective action intentions, these paths were largely non-significant. As can be seen in Supplementary Table S5, changes in people's belief that the movement could be effective predicted increases in their conventional collective action intentions at later time points. However, none of the other longitudinal paths were statistically significant when predicting conventional collective action intentions. In contrast, when we examined the reverse direction—where collective action intentions predicted subsequent change in psychological variables—we observed a number of significant effects. Specifically, increases in conventional collective action intentions were associated with increases in anger, moral conviction, identification, efficacy, and collective victimhood.

Interestingly, none of the longitudinal analyses involving radical collective action intentions were significant (see Supplementary Table S6). As shown in our cloud plots (Fig. S3) and descriptive statistics (Table S4), radical intentions displayed an extreme floor effect, with the majority of respondents scoring below '2' on a 1–7 scale and only a very small

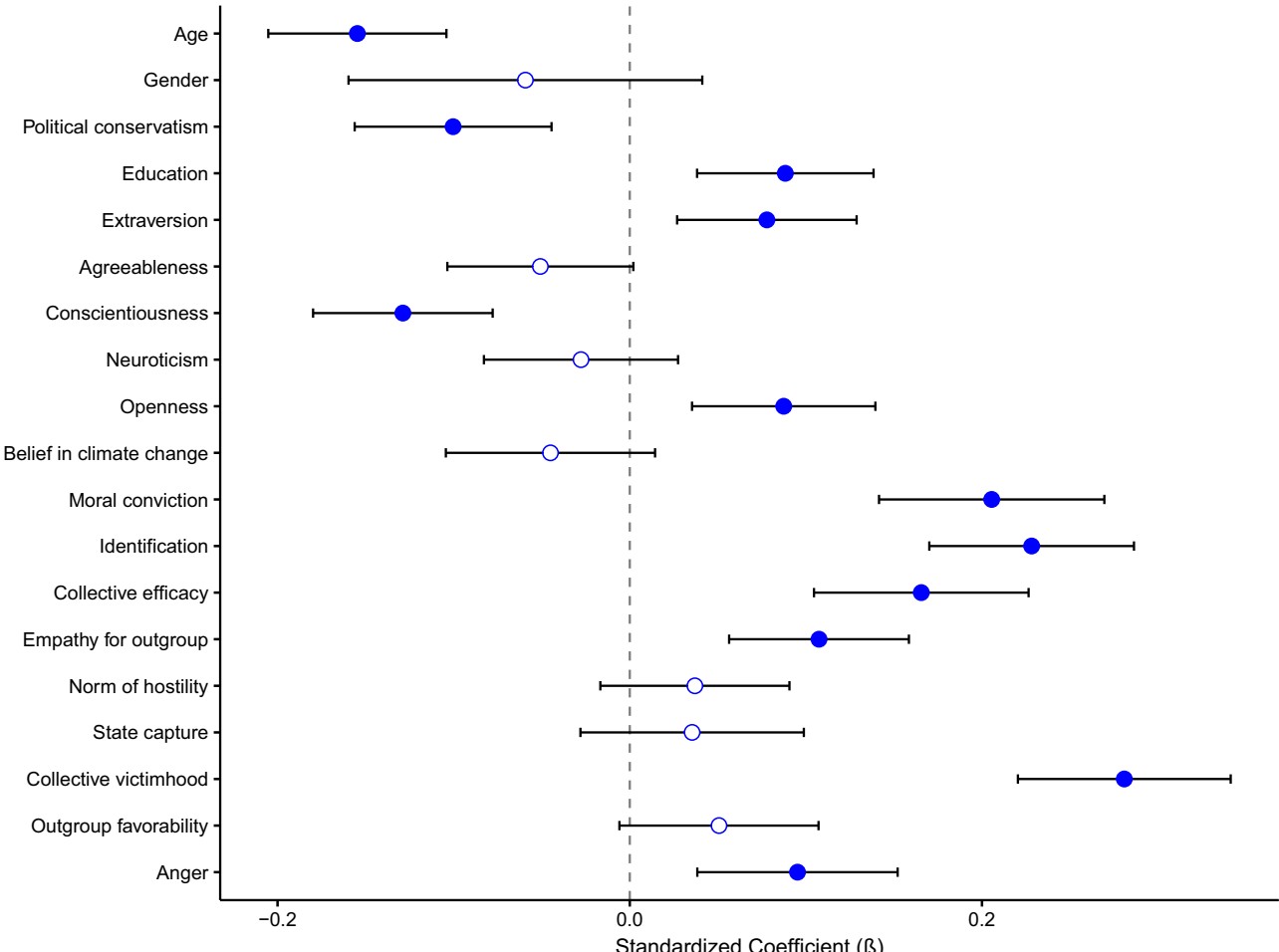

**Fig. 1 | Predictors of support for conventional collective action as analysed over time (through multi-level modelling).** Error bars are 95% confidence intervals. Shaded in diamonds represent significant unique prediction. *N* = 1427 participants.

proportion endorsing these behaviours. This restricted variance limits the inferences that can be drawn about predictors of radical intentions, and it likely contributes to the absence of longitudinal effects in the DSEM models. Radical climate action is a genuinely low-base-rate phenomenon, and rare behaviours measured via self-reported intentions are particularly susceptible to distributional constraints. As such, the DSEM null findings should not be framed as evidence against within-person effects but may in fact reflect limited variance over time, floor effects, and the present-day reality that radical intentions among supporters of climate action are fairly stable, low-frequency orientations.

## Discussion

The goal of this paper is to examine the extent of intentions to engage in radical collective action on climate change, as well as the predictors of those intentions, a question that is under-researched and under-theorised. Among our sample of 1427 supporters of climate change action, only 3.9% scored above the midpoint of the scale, and nearly half the sample recorded scores of 1 (strongly disagree) for all radical action statements. These Australian data converge with research showing low levels of positivity toward non-violent but disruptive climate action tactics in a U.S. sample[67]. Low intentions to engage in radical action did not reflect a generalised apathy or aversion to collective action: over a quarter of the sample scored above the midpoint on the scale of conventional collective action intentions.

More relevant to the current paper are the psychological and demographic predictors of radical climate action intentions. Here, there were four surprises. First, those high in radical action intentions did not appear to be especially left-wing. Second, those high in radical action intentions did not

appear to have especially strong belief in climate change. Indeed, after controlling for the other predictors, radical action intentions were *negatively* associated with belief in anthropogenic climate change. To be clear, supporters of radical action were not sceptics; belief in anthropogenic climate change was uniformly high in our sample. However, the patterns of correlation underscore the extent to which the profile of radical actors in our sample deviates from the 'left-wing green ideologue' stereotype. This did not appear to be a result of measurement error because the same measures (of political conservatism and belief in climate change) shared predictable relationships with conventional collective action intentions. Interestingly, a parallel to this finding was recently reported in correlational work on 293 White Britons: perceptions of the justifiability of sabotaging infrastructure and physically harming leaders of big carbon-emitting industries were *positively* correlated with belief that climate change is a hoax[44].

The third surprise was that, although theorists have advanced the notion that radical collective action would be uniquely associated with outgroup antipathy (dehumanisation of the perceived oppressor), there was limited evidence for this in our sample of ordinary community members. Indeed, those who scored relatively high on the radical action scale expressed greater favourability towards opponents of climate change action and more empathy toward opponents.

Fourth, despite theorising linking radical action intentions to perceptions of efficacy, no reliable association was found in our data. This contrasts with the findings for conventional collective action intentions, where efficacy was the only variable to predict both cross-sectionally and longitudinally. Notably, our measure captured *collective efficacy* (beliefs about what people concerned about climate change can achieve together) rather

**Table 2 | Multilevel model results predicting conventional collective action intentions over 1 year**

| Predictor | β | 95% CI | SE | df | t | p |
|---|---|---|---|---|---|---|
| Age | **−0.15** | [−0.205, −0.104] | **0.03** | **1417.53** | **−6.00** | **<0.001** |
| Gender | −0.06 | [−0.159, 0.040] | 0.05 | 1375.01 | −1.16 | 0.247 |
| Political conservatism | **−0.10** | [−0.156, −0.045] | **0.03** | **1372.12** | **−3.52** | **<0.001** |
| Education | **0.09** | [0.039, 0.138] | **0.03** | **1342.74** | **−3.46** | **<0.001** |
| Extraversion | **0.08** | [0.027, 0.128] | **0.03** | **1356.14** | **2.30** | **0.003** |
| Agreeableness | −0.05 | [−0.103, 0.002] | 0.03 | 1336.51 | −1.89 | 0.059 |
| Conscientiousness | **−0.13** | [−0.179, −0.078] | **0.03** | **1395.77** | **−4.96** | **<0.001** |
| Neuroticism | −0.03 | [−0.082, 0.027] | 0.03 | 1320.71 | −0.99 | 0.324 |
| Openness | **0.09** | [0.036, 0.139] | **0.03** | **1343.22** | **3.29** | **0.001** |
| Belief in climate change | −0.05 | [−0.104, 0.014] | 0.03 | 1448.06 | −1.49 | 0.137 |
| Moral conviction | **0.21** | [0.142, 0.269] | **0.04** | **1416.00** | **6.30** | **<0.001** |
| Identification | **0.23** | [0.171, 0.286] | **0.03** | **1479.33** | **7.70** | **<0.001** |
| Collective efficacy | **0.17** | [0.105, 0.226] | **0.03** | **1417.81** | **5.33** | **<0.001** |
| Empathy for outgroup | **0.11** | [0.057, 0.158] | **0.03** | **1423.67** | **4.13** | **<0.001** |
| Norm of hostility | 0.04 | [−0.016, 0.090] | 0.03 | 1458.60 | 1.35 | 0.177 |
| State capture | 0.04 | [−0.028, 0.098] | 0.03 | 1431.34 | 1.10 | 0.273 |
| Collective victimhood | **0.28** | [0.221, 0.341] | **0.03** | **1446.70** | **9.12** | **<0.001** |
| Outgroup favourability | 0.05 | [−0.005, 0.107] | 0.03 | 1320.40 | 1.76 | 0.079 |
| Anger | **0.10** | [0.039, 0.152] | **0.03** | **1409.56** | **3.29** | **0.001** |

Gender is coded such that 1 = male, 2 = female. SE = standard error. Beta = standardised effect size.
All the significance levels, including the p values, are in bold.

**Table 3 | Multilevel model results predicting radical collective action intentions over 1 year**

| Predictor | β | 95% CI | SE | df | t | p |
|---|---|---|---|---|---|---|
| Age | **−0.19** | [−0.233, −0.144] | **0.02** | **1388.94** | **−8.22** | **<0.001** |
| Gender | **−0.12** | [−0.204, −0.027] | **0.05** | **1341.81** | **−2.53** | **0.011** |
| Political conservatism | 0.03 | [−0.018, 0.081] | 0.03 | 1342.09 | 1.24 | 0.216 |
| Education | 0.04 | [−0.001, 0.087] | 0.02 | 1310.85 | 1.89 | 0.059 |
| Extraversion | 0.04 | [−0.004, 0.085] | 0.02 | 1325.10 | 1.75 | 0.080 |
| Agreeableness | **−0.07** | [−0.115, −0.022] | **0.02** | **1304.14** | **−2.87** | **0.004** |
| Conscientiousness | **−0.16** | [−0.210, −0.120] | **0.02** | **1367.58** | **−7.14** | **<0.001** |
| Neuroticism | −0.04 | [−0.093, 0.004] | 0.02 | 1287.43 | −1.80 | 0.072 |
| Openness | −0.01 | [−0.051, 0.041] | 0.02 | 1311.67 | −0.23 | 0.820 |
| Belief in climate change | **−0.23** | [−0.281, −0.176] | **0.03** | **1423.99** | **−8.50** | **<0.001** |
| Moral conviction | **0.06** | [0.008, 0.121] | **0.03** | **1388.89** | **2.22** | **0.027** |
| Identification | **0.12** | [0.064, 0.166] | **0.03** | **1457.19** | **4.37** | **<0.001** |
| Collective efficacy | 0.01 | [−0.042, 0.066] | 0.03 | 1391.07 | 0.44 | 0.658 |
| Empathy for outgroup | **0.08** | [0.032, 0.122] | **0.02** | **1397.03** | **3.32** | **<0.001** |
| Norm of hostility | 0.03 | [−0.020, 0.075] | 0.02 | 1434.87 | 1.14 | 0.253 |
| State capture | 0.01 | [−0.045, 0.067] | 0.03 | 1405.47 | 0.38 | 0.708 |
| Collective victimhood | **0.30** | [0.244, 0.351] | **0.03** | **1422.02** | **10.88** | **<0.001** |
| Outgroup favourability | **0.12** | [0.068, 0.167] | **0.03** | **1287.08** | **4.59** | **<0.001** |
| Anger | **0.06** | [0.009, 0.110] | **0.03** | **1382.16** | **2.31** | **0.021** |

Gender is coded such that 1 = male, 2 = female. SE = standard error. Beta = standardised effect size.
All the significance levels, including the p values, are in bold.

than contributive or self-efficacy (beliefs about the impact of one's own involvement). It is possible that different forms of efficacy may be more strongly implicated in willingness to engage in disruptive or radical forms of action, a question that future studies could address directly.

A similar point applies to social identification. In the present study, we used a widely validated single-item measure capturing general identification with others who share one's climate change views. However, recent theoretical developments distinguish between different forms of identification—such as politicised identity, moralised identity, and identity fusion—which may have distinct implications for normative versus non-normative collective action[68,69]. Whereas broad identification typically predicts conventional engagement, more intense or fused forms of attachment have been

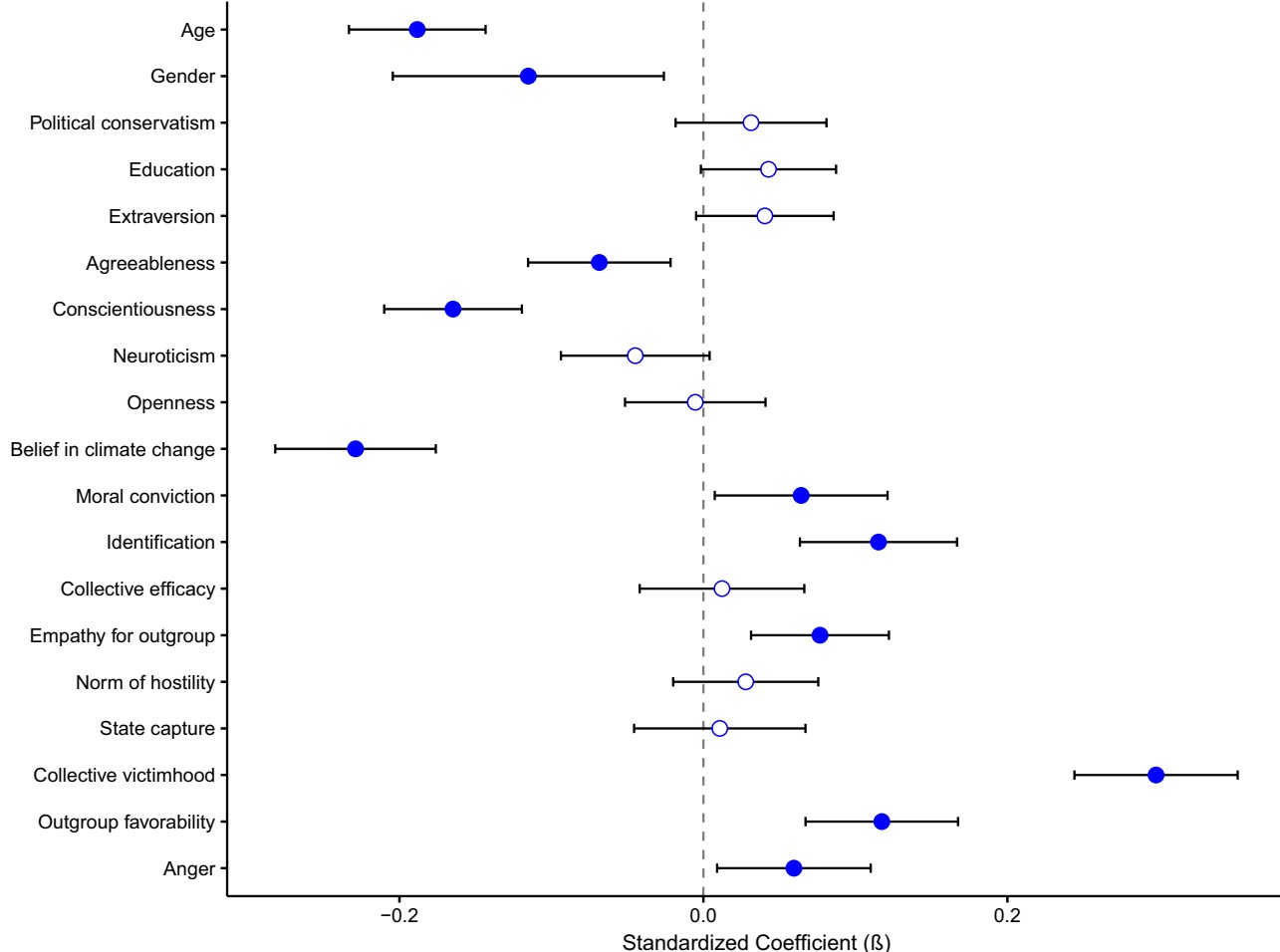

**Fig. 2 | Predictors of support for radical collective action as analysed over time (through multi-level modelling).** Error bars are 95% confidence intervals. Shaded in diamonds represent a significant unique prediction. $N = 1427$ participants.

linked in some studies to willingness to engage in high-risk or costly collective behaviour. Future research could therefore test whether specific types of efficacy and identification, rather than general measures of these constructs, differentially predict the more disruptive or confrontational behaviours that constitute radical action. By incorporating these constructs, our study aligns with contemporary debates emphasising that different types of identification, efficacy, and grievance-based emotions may give rise to distinct forms of collective action rather than a single escalation continuum.

In sum, our data suggest it would not be easy to identify supporters of radical action by knowing whether they feel antipathy toward opponents, the extent to which they feel collective efficacy about climate change, or even by knowing the strength and conviction with which they believe in climate change. So, what are the factors that predict radical climate action intentions? First, those who intend to engage in radical action are younger. It is tempting to attribute this to a 'Greta Thunberg effect' whereby high school students were acculturated into a norm of civil disobedience through the Fridays for Future movement. However, it should be noted that our sample was relatively representative of age, and the average age of the people who scored above the midpoint on radical action was 39.9 years. Rather, the age effect might be better interpreted as part of a broader tendency for younger people to endorse more radical action in political matters[70].

Second, those who intend to engage in radical collective action are somewhat less agreeable and conscientious. Conscientiousness was a particularly strong and resilient predictor of radical action intentions in our sample. Although conscientiousness has previously been established as a positive predictor of motivation to mitigate climate change[61], this is now the second dataset to record a negative association between conscientiousness

and radical climate collective action intentions[45]. It is perhaps not surprising that support for law-breaking or violence should be negatively associated with a personality domain that prioritises orderliness, dutifulness, self-discipline, and cautiousness.

Third, the strongest psychological predictor of all—the variable that out-performed the other predictors in both waves of analysis, and for both conventional and radical action—is collective victimhood. Grounded in literature typically applied to entrenched political conflicts[71–73], this measure captured the extent to which people who support action on climate change perceive having experienced greater harm, trauma, and need for protection compared to those who oppose such actions. The most theoretically interesting feature of this construct is the fact that the causal relationships have been proposed to go multiple ways. On one hand, it is possible that collective victimhood is driving collective action intentions because it reinforces perceptions of injustice, moral urgency, and the need for disruptive resistance[74]. Alternatively, collective victimhood may serve a function of justifying radical tactics. For example, if climate activists perceive that their movement has suffered extreme injustice (e.g. government inaction, fossil fuel lobbying, repression of protests), they may see themselves as victims and justify more radical tactics as necessary in self-defence[75].

Our longitudinal analyses on radical action intentions are inconclusive on this front, possibly reflecting the near-floor effects on radical intentions. However, analyses on conventional collective action demonstrate that engaging with, or even intending to engage with, collective action may itself reshape psychological orientations (on collective victimhood, but also on anger, efficacy, identification and moral conviction), rather than such orientations consistently driving future shifts in action. This finding is

**Table 4 | Comparisons between those who scored above the mid-point on the support for radical action scale and those who did not**

| Variables | Total sample | | Don't support radical action | | Support radical action | | | | |
|---|---|---|---|---|---|---|---|---|---|
| | **M (SD)** | 95% CI | **M (SD)** | 95% CI | **M (SD)** | 95% CI | df | t | p |
| Age | **51.45 (16.96)** | [50.50, 52.41] | **52.01 (16.96)** | [51.08, 52.94] | **39.85 (12.59)** | [36.38, 43.32] | 60.1 | **5.16** | **<0.001** |
| Gender | **1.53 (0.52)** | [1.50, 1.56] | **1.53 (0.50)** | [1.51, 1.56] | **1.36 (0.49)** | [1.25, 1.53] | 59.5 | **2.36** | **0.019** |
| Political conservatism | 3.55 (1.44) | [3.47, 3.62] | 3.53 (1.43) | [3.45, 3.61] | 3.73 (1.48) | [3.33, 4.13] | 59.3 | −1.03 | 0.301 |
| Education | 3.89 (1.38) | [3.81, 3.96] | 3.88 (1.38) | [3.81, 3.96] | 3.91 (1.44) | [3.52, 4.30] | 59.2 | −0.15 | 0.882 |
| Extraversion | **2.79 (1.02)** | [2.74, 2.84] | **2.78 (1.03)** | [2.72, 2.83] | **3.07 (0.77)** | [2.86, 3.28] | 63.4 | **−2.10** | **0.036** |
| Agreeableness | **3.71 (0.80)** | [3.67, 3.75] | **3.73 (0.80)** | [3.69, 3.77] | **3.35 (0.69)** | [3.16, 3.53] | 61.2 | **3.51** | **<0.001** |
| Conscientiousness | **4.20 (0.74)** | [4.16, 4.23] | **4.22 (0.72)** | [4.18, 4.26] | **3.64 (0.81)** | [3.43, 3.86] | 58.7 | **5.83** | **<0.001** |
| Neuroticism | 2.44 (0.94) | [2.39, 2.48] | 2.43 (0.95) | [2.38, 2.48] | 2.66 (0.75) | [2.46, 2.86] | 62.6 | −1.82 | 0.068 |
| Openness | 3.58 (0.77) | [3.54, 3.62] | 3.59 (0.77) | [3.55, 3.63] | 3.46 (0.77) | [3.26, 3.67] | 59.7 | 1.16 | 0.247 |
| Belief in climate change | **6.11 (1.03)** | [6.06, 6.17] | **6.14 (1.01)** | [6.09, 6.20] | **5.50 (1.15)** | [5.19, 5.81] | 58.5 | **4.63** | **<0.001** |
| Moral conviction | **4.85 (1.28)** | [4.78, 4.91] | **4.83 (1.29)** | [4.76, 4.90] | **5.28 (0.96)** | [5.02, 5.54] | 63.5 | **−2.59** | **0.010** |
| Identification | **5.19 (1.23)** | [5.13, 5.26] | **5.18 (1.24)** | [5.11, 5.24] | **5.54 (1.03)** | [5.26, 5.81] | 61.7 | **−2.13** | **0.033** |
| Collective efficacy | 4.93 (1.24) | [4.86, 4.99] | 4.92 (1.25) | [4.86, 4.99] | 5.05 (1.06) | [4.76, 5.33] | 61.4 | −0.72 | 0.470 |
| Empathy for outgroup | **3.77 (1.37)** | [3.70, 3.84] | **3.75 (1.37)** | [3.68, 3.82] | **4.23 (1.34)** | [3.87, 4.59] | 59.8 | **−2.59** | **0.010** |
| Norm of hostility | **4.44 (1.62)** | [4.35, 4.52] | **4.41 (1.64)** | [4.33, 4.50] | **5.02 (1.04)** | [4.74, 5.30] | 66.7 | **−2.74** | **0.006** |
| State capture | 5.21 (1.40) | [5.13, 5.28] | 5.20 (1.42) | [5.13, 5.28] | 5.35 (1.02) | [5.07, 5.62] | 62.7 | −0.74 | 0.461 |
| Collective victimhood | **4.04 (1.43)** | [3.96, 4.11] | **3.99 (1.43)** | [3.92, 4.07] | **5.11 (0.94)** | [4.85, 5.36] | 65.8 | **−5.78** | **<0.001** |
| Outgroup favourability | **37.84 (20.61)** | [36.77, 38.91] | **37.41 (20.25)** | [36.33, 38.49] | **48.28 (26.62)** | [41.01, 55.54] | 55.5 | **−3.82** | **<0.001** |
| Anger | 3.97 (1.55) | [3.89, 4.05] | 3.95 (1.55) | [3.87, 4.04] | 4.34 (1.61) | [3.91, 4.77] | 59.3 | −1.82 | 0.069 |

All continuous variables were measured on 1–7 scales except age, education (1–6) and outgroup favourability (0–100). Gender was coded such that 1 = male, 2 = female. Columns refer to respondents on or below the midpoint of the radical action scale ('don't support radical action') versus those above the midpoint of the radical action scale ('support radical action').
All the significance levels, including the p values, are in bold.

consistent with dynamic models of collective action that emphasise reciprocal influences between action and psychology[6,23,68]. Rather than viewing psychological predictors as static antecedents, our results highlight the possibility that intentions to act can play a formative role in consolidating identities, reinforcing efficacy and shaping attitudes over time.

**Limitations**

Although the multilevel models identified several variables associated with higher radical action intentions, these findings must be interpreted cautiously. Radical intentions were extremely low and highly skewed across the sample, producing restricted variance that necessarily constrains the precision of linear estimates. The significant predictors we observed—particularly collective victimhood and lower conscientiousness—showed convergence across analytic approaches, but these associations should be understood as preliminary rather than definitive.

It should also be acknowledged that the processes shaping the radical action intentions of our large sample of ordinary citizens would not be expected to generalise to members of specific organised groups such as Extinction Rebellion. Indeed, one empirical study that focused on Extinction Rebellion members reached conclusions that converged closely with SIMCA (whereby radical action was primarily predicted by moral conviction, collective efficacy, anger and identification)[41]. This discrepancy parallels the literature on violent extremism, where group processes, identities, and norms are more at play in predicting group action than lone actor attacks[76,77]. But it also reflects a broadening of the theoretical lens to incorporate the psychology of ordinary community members who care about climate change but might not self-report as activist or be part of an organised environmental group.

While intentions offer a proximal indicator of future behaviour, they do not always translate into action due to barriers such as opportunity, efficacy and resource constraints[78,79]. Thus, future research should examine whether the predictors identified here extend to observed participation in

climate-related collective action. Moreover, intentions may not fully capture the situational and group-level processes (e.g. deindividuation, crowd dynamics and heightened arousal) that sometimes are associated with radical behaviour. These processes are unlikely to be reflected in stable individual differences measured via a panel survey, and may account for additional variance in real-world radical action beyond what can be modelled here.

Our findings should be considered in light of the Australian context in which the study was conducted, but also with a view to possible broader generalisation. Australia represents a sociopolitical environment characterised by high public awareness of climate change but historically polarised political leadership and policy inconsistency. These contextual dynamics may influence both the strength and nature of the psychological processes examined here. However, awareness of climate change and polarised political leadership is characteristic of numerous Western countries, and also some non-Western ones. Future research would benefit from testing whether the patterns observed replicate across different geopolitical and cultural settings.

Another limitation of the present study is that we did not measure participants' prior political or activist engagement, which the literature suggests may be an important factor shaping support and participation[80]. Future research should incorporate this dimension to provide a fuller picture of mobilisation processes. Future research could also examine how justice orientations and conflict narratives shape responses to climate action. For instance, perceptions of collective victimhood may be linked with retributive frames that justify punitive or radical tactics, whereas restorative frames may channel support toward more conventional, cooperative action[81]. Finally, although the longitudinal design employed here strengthens inference relative to single-wave cross-sectional research, future studies using experimental manipulations or activist samples would be valuable in triangulating and extending the pathways identified in the present work.

## Conclusions

This study sheds light on the psychological and demographic predictors of radical climate action intentions, challenging prevailing assumptions about who endorses such tactics and why. Contrary to expectations, radical climate action intentions were not associated with collective efficacy, political ideology, or strong belief in climate change. Instead, intentions to engage in radical action were best predicted by youth and personality. Most notably, collective victimhood emerged as the strongest predictor of both conventional and radical action intentions, suggesting that perceived suffering and injustice either cause or are invoked by the full spectrum of climate action tactics.

These findings have important theoretical and practical implications. Theoretically, they indicate that existing models of collective action, developed primarily for conventional protest, may not fully account for the unique motivations behind radical climate activism. The role of perceived victimhood especially warrants further exploration, as it may serve to both motivate activism and justify more extreme tactics. Practically, understanding the psychological underpinnings of radical climate action has implications for policymakers, law enforcement and activist groups. If radical tactics alienate broader public support, movements may need to understand why they are drawn to it, and reconsider their approaches[13,82]. Moving forward, future research should explore how perceptions of injustice and intergroup dynamics shape the evolving landscape of climate activism and what this means for the broader push toward climate mitigation.

## Data availability

Data are available on OSF: https://doi.org/10.17605/OSF.IO/8D9NH.

## Code availability

Code is available on OSF: https://doi.org/10.17605/OSF.IO/8D9NH.

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

## Acknowledgements

Participant compensation and salaries of some authors were supported by the Australian Research Council Discovery grant scheme, DP220101566, Australian Research Council Fellowship scheme, FT240100558, and Australian Research Council Laureate Fellowship, FL230100022. The funders had no role in study design, data collection and analysis, decision to publish or preparation of the manuscript.

## Author contributions

Matthew Hornsey: conceptualisation; methodology; validation; formal analysis; writing—original draft; supervision; funding acquisition. Samuel Pearson: software; validation; formal analysis; visualisation; writing—review & editing. Susilo Wibisono: methodology; investigation; data curation; writing—review & editing; project administration. Emma Thomas: conceptualisation; methodology; validation; formal analysis; writing—review & editing; supervision; funding acquisition. Lucy Bird: validation; formal analysis; writing—review & editing. Jarren Nylund: resources; visualisation; writing—review & editing. Chris Bretter: formal analysis; visualisation; writing—review & editing. Janquel Acevedo: visualisation; writing—review & editing. Kelly Fielding: writing—review & editing; funding acquisition. Catherine Amiot: writing—review & editing; funding acquisition. Fathali Moghaddam: writing—review & editing; funding acquisition. Winnifred Louis: conceptualisation; methodology; writing—review & editing; supervision; funding acquisition.

## Competing interests

The authors declare no competing interests.
