## [Transparent Peer Review File · Communications Psychology]

Youth, personality and collective victimhood distinguish support for radical climate action

Corresponding Author: Professor Matthew Hornsey

Version 0:

Decision Letter:

Dear Matt,

Thank you for your patience during the peer-review process. Your manuscript titled "Rethinking the psychology of support for radical climate activism" has now been seen by 2 reviewers, whose comments are appended below. You will see that they find your work of some potential interest. However, they have raised quite substantial concerns that must be addressed. In light of these comments, we cannot accept the manuscript for publication, but would be interested in considering a revised version that fully addresses these serious concerns.

We hope you will find the Reviewers' comments useful as you decide how to proceed. Should additional work allow you to address these criticisms, we would be happy to look at a substantially revised manuscript. If you choose to take up this option, please highlight all changes in the manuscript text file, and provide a detailed point-by-point reply to the reviewers.

Editorially, we consider it critical that the revision analyses the longitudinal data and reports all results, including the intermediate survey responses.

I am attaching a checklist that details critical reporting requirements for the revised manuscript. Please attend to each item and ensure your manuscript is fully compliant. We are requesting that your manuscript aligns with these requirements as this facilitates the evaluation of your manuscript, reducing delays in re-review and potential future acceptance. If your revised manuscript is not aligned with these requests on major issues, such as those concerning statistics, it may be returned to you for further revisions without re-review. Additional information can be found in our style and formatting guide Communications Psychology formatting guide.

If the revision process takes significantly longer than five months, we will be happy to reconsider your paper at a later date, provided it still presents a significant contribution to the literature at that stage.

Please use the following link to submit your

- revised manuscript,
- point-by-point response to the referees' comments,
- cover letter (as a separate document),
- the Editorial Policy Checklist (see below),
- the Reporting Summary (see below), and

- the completed Editorial Request Table (attached):

Link Redacted

Thank you for the opportunity to review your work.

Best regards,

Marike

Marike Schiffer, PhD
Chief Editor
Communications Psychology

REVIEWER REPORTS:

Reviewer #1 (Remarks to the Author):

Thank you for the opportunity to review this work. I appreciate the authors' meticulous effort in addressing this timely subject. However, I have several substantial concerns regarding the study's rationale and conceptualization:

The first issue is the conflation of collective action support and participation. Throughout the manuscript (particularly in the introduction and discussion), support for collective action and participation in collective action are discussed interchangeably. However, the theories examined in the introduction primarily focus on participation rather than community support. Several studies in the literature have argued that the social psychological processes underlying participation and support differ significantly, as do their predictors. This conflation weakens the rationale for the selection of specific predictors over others. Furthermore, since the introduction provides a broad overview rather than a well-justified and specified arguments & summary, the chosen predictors appear somewhat arbitrary.

Second, one of the central arguments of the paper is that climate collective action is a qualitatively distinct form of collective action due to the complexity and amorphous nature of intergroup contexts. However, this claim is not sufficiently justified. In lots of climate-related mobilisations—particularly radical climate activism—ingroup-outgroup distinctions are often clearly defined and historically informed (e.g., protesters vs. law enforcement; see Uysal et al., 2025). Additionally, the notion of fluid and complex intergroup boundaries is not unique to climate activism. Most social movements involve complex and intersecting ingroup-outgroup dynamics. Black liberation movement is not simply about Black vs. White identities, nor is feminism solely about women vs. men. Many movements involve multiple fronts, including clear ingroups, outgroups, allies, potential allies, and third parties. The argument regarding the distinctiveness of climate collective action requires further justification or revision.

Third, some claims in the introduction and conclusions in the discussion appear decontextualised. The fact that the data were collected from Australian participants is not properly mentioned until the method section. This omission gives the impression that the geographic and sociopolitical context has no bearing on the study's conclusions or arguments. Greater acknowledgement of how the Australian context might shape the findings is crucial.

Fourth, the authors present several distinct theoretical perspectives under the broad umbrella of SIMCA (Social Identity Model of Collective Action) and its extensions. However, this approach risks being reductive, particularly in the case of DIME and MOBILISE. These models do not merely extend or elaborate on SIMCA; they also explore the dynamic processes that sustain mobilization. While they are influenced by SIMCA, they incorporate broader mechanisms beyond psychological predictors, which should be acknowledged.

Finally, throughout the manuscript, several claims require additional citation support. Some statements appear speculative or broad, and reinforcing them with relevant literature would strengthen the argumentation.

Reviewer #2 (Remarks to the Author):

The present article "Rethinking the psychology of support for radical climate activism" presents data from a cross-sectional sample of the Australian public that supports climate action in an attempt to test how traditional theories of collective action (e.g., SIMCA) align with normative and non-normative 'radical' action. Specifically, the article attempts to study this in the domain of climate change, where ingroup and outgroup identification becomes difficult to disentangle, resulting in questions about the validity of these theoretical constructs in this domain. While I think the questions raised in this piece are worthwhile

areas of study, I do have concerns about the sample, conceptualization, and analysis. Many of these can be addressed through edits to the current manuscript, including additional details/analyses. However, to answer the questions posed in this manuscript, I think additional data may be required.

More information needed about the data

I appreciate the descriptive statistics that the authors present throughout the manuscript, that characterize things like overall support for normative and radical collective action. Additionally, the authors present the alpha and correlation coefficients for all scales in supplement, which is helpful to the reader to understand these measures. However, I would request that they also report the mean and SD so readers can evaluate these measures more. Particularly as some may have more limited or skewed distribution and possible ceiling effects.

Also, there is a lack of data visualization that seems key in a cross-sectional analysis like this. We're shown group differences and regression coefficients. But the distribution of variables (particularly the support variables) is important to see. This may be particularly important for some associations, such as belief in climate change and support for radical climate action. Showing histograms, scatterplots, violin plots, etc. can help illustrate the nature of the relationship, and allow for critical analysis of if there are issues in the data (e.g., extreme outliers).

One final note about additional information needed from the data, is that there should be some additional information about support for normative and radical action. These are the key outcomes of this study, and are treated only as composite measures. While both have very high internal reliability, I think two types of information would be helpful in evaluating them. First, factor analysis to show a) that these are two distinct composite measures and b) that there aren't subfactors or groupings of types of action within each. Second, particularly for the normative collective action, people may support some actions more than others. Descriptive information for each item could be useful for the reader and add another layer of detail to your arguments (though this does not need to be in the main text).

Concerns about measurement and conceptualization

The authors made a good effort to include many relevant factors in their analysis. However, I think there are some issues with their conceptualization and/or measurement tools used. These issues may not invalidate their findings but gives several caveats to the conclusions they can draw.

First, the primary outcome measures are consistently referred to as 'support'. However, these are behavioral intentions. This is an issue for two reasons. First, intentions require intentions to act, which relies on many other factors (e.g., people may feel that they don't know how or where to start, what to do, that they cannot physically attend protests or engage in violent protest). So, the predictive factors here that should lead to action may be mitigated by external factors, leading the intention-behavior gap to be quite large. For example, there was a good amount of 'support' for normative collective actions. However, only a minority of people regularly (if ever) engage in actual protests. This could mean that, while the factors in the model predict intentions, they may not actually predict behavior. Second, support is actually an important part of protest, in that bystanders and the general population's approval of collective action is an important part of creating change. As such, I would urge the authors not to conflate support and intentions.

Secondly, identification is a key part of many previous models (e.g., SIMCA, SIMPEA), so its inclusion is well warranted. However, there are a few issues with how it is measured. First, the measurement tool is mostly reverse scored items, so is it a measure of identification or a measure of social distance from opponents of the movement? It also is not identification with a specific movement but with 'other people who have similar views on climate change as you do' (whatever those views are). This means that it may be either a measure of identification with the broad swath of the population that supports climate action in principle (an overly broad identification measure that may not be particularly motivating) or disidentification with the opposition. Given this, it's not unsurprising that it does not show significant associations with radical action. Particularly as radical or progressive actors may already distance themselves from the rest of the general pro-climate public (see the vegan vs. vegetarian dynamic), but instead identify with other, similar, radical climate actors. I agree that an ingroup-outgroup dynamic is difficult to concisely conceptualize in this domain, but it is hard to say that ingroup matters or doesn't when it's such a broad definition.

I also would appreciate more discussion on efficacy specifically. There are many forms of efficacy (self, collective, contributive). The form of efficacy measured here does not relate to radical action intentions. However, that isn't surprising if radical actions are seen as an individual action meant to raise awareness or cause harm rather than solve the problem directly.

Analyses

I appreciate the use of alternative analysis methods. However, the paper feels fairly brief as is, and the re-analysis (multilevel model vs. regression) does not seem to add new content. I might suggest that the authors consider presenting only one of those types of models in the main text, and reporting the rest in supplement (however this is not a critical issue).

I would however suggest an additional analysis for the authorial team to present. The multilevel model currently treats support at the different time points simply as indicators to an underlying latent trait of 'willingness to take action' across time. However, this dataset gives the authors a chance to actually analyze these results longitudinally, making their contribution much more significant. If there is change in support across the timepoints of the study, treating that at a linear model and

using the various measures to predict changes in support would seem to be a greater indicator of actual intention/behavior shift. Alternatively, if that model does not converge, using difference scores from time 1 to time 2.

Regardless of whether it is analyzed in a longitudinal fashion or not, I do not think it is acceptable to not present the intermediate survey results. Assuming there is no change in the pattern of findings, then there's no reason to omit them. And if there is a change, then it becomes especially critical to show those effects. Omitting reporting this time point presents concerns about cherry-picking ideal contrasts across specific times points. If the intermediate set of data didn't contain all variables, you should use only those present or use the predictors from time 1. Also, the argument that there are high levels of noise is unclear. How was this assessed? I suggest presenting these results, and if you want to make an argument about statistical noise, present the data that shows that they should be omitted. This can all be presented in supplement, but it shouldn't be omitted.

On a larger point though, even with these alternative analyses, the issue is still cross-sectional, examining intentions in a broad section of the Australian public that may never actually follow-up on these intentions. As such, this one study is hard to draw conclusions from. My suggestion to the authors would be to add to the study, adding further evidence that supports their findings of what motivates radical and normative climate action (e.g., experimental data that manipulates the theorized constructs that motivate radical action, cross-sectional analysis on actual climate activists, qualitative data, etc.)

Miscellaneous points

In the abstract, the authors say that "These findings challenge media portrayals of radical climate activists". What is meant by this?

The authors say that there is moderate support, but only 25% of the sample were above the midpoint. This seems like a poor way to describe as opposed to moderate or broad opposition.

There was no measure of prior political or activist engagement? That seems like a critical variable to omit given the past literature on the important role of attending events and engaging with activists in actually mobilizing action.

"Interestingly, perceptions that other supporters felt hostility toward opponents reliably predicted radical collective action support, even when respondents' own feelings of hostility did not." But you don't measure hostility. If you're referring to things like empathy, that's a different measure.

Please report average political leaning after exclusions (rows 431 to 434)

Only some of the data and script are shown in the OSF link. None of the MPlus script is there.

For all supplemental tables, it would be helpful to bold the significant effects, as you did for Table S7.

One point that might be missed in radical versus collective action is the call for retributive vs. restorative justice, in regards to outgroups. If the outgroup is someone who caused the issue or proved to be an obstacle to solving it, should beliefs about punishment as justice predict willingness to take violent (versus peaceful) action?

As you're looking at conflict literature and applying it to this area of work, I would suggest 'conflict narratives' as a potential future direction or other factor relevant in this work. As the climate movement has been ongoing for decades, people may have developed narratives about the nature of the conflict that can give them meaning in the conflict and fuel radicalization. (e.g., Ulug et al., 2021, How do conflict narratives shape conflict- and peace-related outcomes among majority group members).

In total, I like this paper and the arguments it is trying to make. The difference between normative and non-normative (radical) action is a topic that, while previously explored, is still relatively omitted from psychological literature. Expanding existing theories to address this distinction is key, particularly in the domain of climate action where intergroup lines and dimensions can become nebulous. This is a critical area of exploration for our field. However, I have reservations about the manuscript in its current form. To further support the conclusions of this study, I would suggest the inclusion of additional data. However even as is, I think there are some edits that can be made to strengthen the paper, address concerns about the data, and modify the conclusions that can be drawn from the analysis.

EDITORIAL POLICIES

We ask that you ensure your manuscript complies with our editorial policies and reporting requirements.

To that end, we require revised manuscripts to be accompanied by two completed items: a reporting summary that collects information on study design and procedure, and an editorial policy checklist that verifies compliance with all required editorial policies

- <https://www.nature.com/documents/nr-reporting-summary.zip>>Nature Research Reporting Summary
- <https://www.nature.com/documents/nr-editorial-policy-checklist.pdf>>Editorial Policy Checklist

All points on the policy checklist must be addressed. Your revised manuscript can only be sent back to the referees if these checklists are completed and uploaded with the revision.

Notes: If you have submitted a Stage 1 Registered Report, Review, Primer, Comment, or Perspective you do not need to submit these forms. If you have already submitted these forms, you may disregard this request.

If you experience problems in linking your ORCID, please contact the <http://platformsupport.nature.com/>>Platform Support Helpdesk.

Version 1:

Decision Letter:

Dear Matthew,

Thank you for your patience during the peer-review process. Your manuscript titled "Rethinking the psychology of radical climate activism" has now been seen by the two reviewers, and I include their comments at the end of this message. They find your work improved. However, Reviewer #1 lists a number of persisting shortcomings in the conceptual presentation and treatment of the literature, with which Reviewer #2 concurs. However, Reviewer #2 in particular also notes the value of the descriptive findings that emerge from your study and we editorially welcome the recent inclusion of wave 2 data.

While we would be interested in receiving a revised version, I must also highlight that we can only consider the revision any further for eventual publication if these points are suitably addressed.

Regarding the treatment of the literature, we do ask you to ensure that the relevant background literature is fully incorporated, that you clarify how the approach conceptually maps onto what is known and how the work can inform the literature (even in the face of differences in approach). Any issue that can be addressed through control analyses should be approached empirically. You previously included a Table to explicate the predictions you had for the analyses and how these map onto different theories or past findings. I ask that that Table is maintained, but that you highlight that it does not constitute a preregistered set of hypotheses.

As you revise the manuscript in response to these issues, please also implement all requests in the attached Mandatory Revision Requests document. All requirements listed in this document need to be fully met, or the work will be returned to you for further revisions without peer review. This workflow is in place to increase the likelihood that the paper will be accepted for publication. It reduces the number of rounds of revision (and review) and ensures that the reviewers vet a version of the article that is compliant with journal policies. If you have any questions regarding the required revisions, please contact the journal prior to resubmission to avoid a negative outcome.

Please submit the following items:

- Revised manuscript
- Point-by-point response to the referees' comments
- Mandatory Revision Requests Table (attached).
- Cover letter (as a separate document)

via this link: Link Redacted .

** This url links to your confidential home page and associated information about manuscripts you may have submitted or are reviewing for us. If you wish to forward this email to co-authors, please delete the link to your homepage first **

Best regards,

Marike

Marike Schiffer, PhD
Chief Editor
Communications Psychology

REVIEWER REPORTS:

Reviewer #1 (Remarks to the Author):

There are still some substantial issues that remain unaddressed. First, the measurement of collective action combines multiple constructs without sufficient theoretical justification. Some items measure support for collective action, while others assess collective action intentions. The provided factor analysis, in its current form, does not sufficiently warrant this combination. Also, two cross-loading items were not discussed or reflected upon in either the methodology or discussion, which raises concerns about construct validity. There is a clear mismatch between how these constructs are introduced in the introduction and how they are operationalised in the study. For example, some items labelled as "radical collective action" (such as chaining oneself to a fence) are not necessarily law-breaking or violent, depending on the context. Second, the literature review remains rather superficial. In recent years, numerous studies and theoretical discussions have advanced our understanding of radical collective action and resistance. Focusing solely on SIMCA and other mainstream approaches provides only a very general overview. Given the current depth and sophistication of the literature, such a broad and limited review does not adequately situate the study within contemporary theoretical debates. Furthermore, the predictors included in the study are not theoretically introduced or justified. Without a grounded discussion of how and why certain variables might differentially relate to conventional versus radical actions, the theoretical framing remains underdeveloped, and the inclusion of these variables appears somewhat arbitrary. There are lots of new insights and debates in the literature on the role of identity and efficacy (particularly, what kind of identification and what kind of efficacy actually lead to radical over conventional collective action), which authors do not engage in any of these discussions.

Reviewer #2 (Remarks to the Author):

The revisions have largely addressed my previous concerns, and I think the current version of the manuscript could be suitable for publication. However, there are a few issues that still need to be addressed.

First, the provision of the cloud plots for the outcome measures is helpful, but it does further highlight the intensive skew and floor effect on radical action intentions. While that skew is noted in the manuscript, I think there are additional limitations to be noted. First, we can see that a supermajority of responses are below a 2 on the scale. While it's true that an exceedingly

small minority of the public will ever engage in radical, violent protest, this accentuates the issue of the gap between action and intentions. Intentions may be low across the board but noting that other social factors such as deindividuation and mob mentality may be more important in promoting action than reasoned intentions. This skewness can harm what inferences we can make about linear associations between this factor and other variables. And the t-tests try to accommodate for this, but the comparison group (everyone above the midpoint) includes people who still outright disagree with most of these items (i.e., a 2-3 on the scale). As such, it may be worth considering a different cutoff point in relation to the scale's values (average of 3+ for example) to compare outright disapproval to approval. However, this further limits the sample size for these comparisons.

Given that this is the primary takeaway from this paper (i.e., what does or does not predict radical action intention) this is a major issue. Without meaningful variability in this outcome, it is difficult to draw conclusions about predictors/differences. The DSEM model may alleviate this, however, it found no significant predictors at all.

As such, this limitation needs to be noted. The authors currently discuss in the discussion, without hedging their claims, the various associations with radical action intentions, and do not mention this issue of skew or limited variance in their limitation section of the discussion. The only place it is mentioned is as a caveat to their non-significant DSEM findings. As written, this frames that lack of findings as potentially untrustworthy, but their other findings as relatively certain. Discussion of what was found to predict conventional collection action intentions is suitable and warranted, but the other findings should be severely limited.

Second, there is need for additional context in the academic literature in regards to the differences between radical and conventional collective action intentions. The authors mention some other terminology that has been used to describe differences in these types of action (Introduction paragraph 2, lines 37-42) but do not cite past work. For example, they mention 'normative' and 'non-normative' collective action without contextualizing how their concepts are the same or different from this and/or what the findings of those past studies are. For example, Selvanathan & Leidner, 2020) found different effects of attachment and glorification on normative and non-normative collective action support, mediated by different paths of retributive versus restorative justice. As another example, members of the lay public had different perceptions of 'typical' and 'atypical' environmentalists, resulting in differences in intentions and support through desired social distance (Bashir et al., 2013). While neither of these papers put forth a specific model (e.g., SIMCA), both clearly have connections to this distinction. Further explanation of what matters in this difference (e.g., legality, normativity, violence, disruption, prototypicality) and how that has been explored in the climate domain and other domains is necessary.

I still think there are some interesting findings to be presented from this paper. For example, even the descriptive findings about levels of intentions for conventional and radical actions is an interesting statistic worth reporting. Similarly, the factor analysis could highlight meaningful differences in understandings/intentions of different types of actions that can further supplement past research and help in situating their findings in past literature that may not have done empirical measurement work to distinguish these types of acts. However, without additional discussion of past findings in the literature we are left without clear guidance on how this fits with other (currently uncited) works. And heavy edits are needed to correctly hedge all claims regarding predictors of radical action intentions given the critical measurement issues.

If you experience problems in linking your ORCID, please contact the Platform Support Helpdesk.

Version 2:

Decision Letter:

Dear Matthew

Your manuscript titled "Youth, personality and collective victimhood distinguish support for radical climate action" has now been seen by our reviewers, whose comments appear below. In light of their advice I am delighted to say that we are happy, in principle, to publish a suitably revised version in *Communications Psychology*.

We therefore invite you to revise your paper one last time to address the remaining concerns of our reviewers and a list of editorial requests. At the same time we ask that you edit your manuscript to comply with our format requirements and to maximise the accessibility and therefore the impact of your work.

EDITORIAL REQUESTS:

SUBMISSION INFORMATION:

OPEN ACCESS:

* **TRANSPARENT PEER REVIEW:** *Communications Psychology* uses a transparent peer review system. On author request, confidential information and data can be removed from the published reviewer reports and rebuttal letters prior to publication. If you are concerned about the release of confidential data, please let us know specifically what information you would like to have removed. Please note that we cannot incorporate redactions for any other reasons.

Link Redacted

Best regards,

Marike

Marike Schiffer, PhD
Chief Editor
Communications Psychology

REVIEWERS' COMMENTS:

Reviewer #1 (Remarks to the Author):

Although the introduction/literature review could have been strengthened further, I don't want to burden authors with another substantial revision round; therefore, I believe my concerns are sufficiently addressed. I only want to raise one minor issue. Please support your claims with citations for the following argument:

The "participate in a sit-in" item loaded primarily on the conventional factor ($\lambda = .58$) but also showed a modest secondary loading on the radical factor ($\lambda = .40$), consistent with the ambiguous status of sit-ins in the literature, where they are

variously classified as conventional, disruptive, or borderline non-normative depending on context.

RESPONSE TO REVIEWERS

Reviewer #1

Thank you for the opportunity to review this work. I appreciate the authors' meticulous effort in addressing this timely subject. However, I have several substantial concerns regarding the study's rationale and conceptualization:

The first issue is the conflation of collective action support and participation. Throughout the manuscript (particularly in the introduction and discussion), support for collective action and participation in collective action are discussed interchangeably. However, the theories examined in the introduction primarily focus on participation rather than community support. Several studies in the literature have argued that the social psychological processes underlying participation and support differ significantly, as do their predictors. This conflation weakens the rationale for the selection of specific predictors over others. Furthermore, since the introduction provides a broad overview rather than a well-justified and specified arguments & summary, the chosen predictors appear somewhat arbitrary.

We appreciate your comment that we should not conflate support with participation. Before responding, we note that Reviewer 2 pointed out that the “support” items nearly all refer to intentions and that we should not conflate support and intentions. As such, we have taken Reviewer 2’s advice to describe the outcome variables as collective action intentions rather than support for collective action.

Although this partly side-steps the critique made above, we agree that many foundational theories in the collective action literature were originally developed to explain participation rather than more abstract constructs such as support and intention. In response, we have clarified our goal in the Introduction to avoid the appearance of conflation.

“We acknowledge that participation in collective action and intentions to engage in collective action are conceptually distinct (as participation typically entails higher levels of personal cost and involvement). Nevertheless, we follow the majority of the literature on this topic in measuring intentions as a reliable (but imperfect) proxy for behaviour. Given that public backing is an important precursor to action and remains comparatively under-theorised, this reflects a deliberate theoretical extrapolation rather than a conflation of constructs.”

We have also added a note in the Discussion acknowledging that intentions do not always translate into behaviour due to behavioural barriers, and that future work is needed to examine the extent to which the predictors identified here forecast actual participation.

We appreciate the reviewer’s concern that our introduction may appear broad and that the set of predictors could seem somewhat arbitrary. Our intent was to balance two competing priorities. On one hand, because this is among the first large-scale studies to examine intentions to engage in radical climate action in the community, we felt a responsibility to the reader to be expansive and transparent in introducing the full set of predictors that have theoretical or empirical precedent. On the other hand, we aimed to tell a coherent story about how these predictors map onto existing collective action theories. We acknowledge that this creates some breadth in the introduction, but we felt it was important at this early stage of inquiry to err on the side of inclusiveness rather than risk omitting potentially important variables.

Second, one of the central arguments of the paper is that climate collective action is a qualitatively distinct form of collective action due to the complexity and amorphous nature of intergroup contexts. However, this claim is not sufficiently justified. In lots of climate-related mobilisations—particularly radical climate activism—ingroup-outgroup distinctions are often clearly defined and historically informed (e.g., protesters vs. law enforcement; see Uysal et al., 2025). Additionally, the notion of fluid and complex intergroup boundaries is not unique to climate activism. Most social movements involve complex and intersecting ingroup-outgroup dynamics. Black liberation movement is not simply about Black vs. White identities, nor is feminism solely about women vs. men. Many movements involve multiple fronts, including clear ingroups, outgroups, allies, potential allies, and third parties. The argument regarding the distinctiveness of climate collective action requires further justification or revision.

We thank the reviewer for this thoughtful point. We agree that complex, intersecting ingroup–outgroup dynamics are not unique to climate activism and characterise many social movements (e.g., Black liberation, feminist, LGBTQ+). Our intention was not to claim that climate collective action uniquely involves fluid boundaries, but rather that such fluidity is arguably more pronounced and central in this field, spanning political, generational, sectoral, national, and ideological fault lines. Consequently, climate movements may lack the clear and historically entrenched ‘ingroup’ vs ‘outgroup’ antagonisms that underpin much theorising around identification-based motivations for participation in collective action. We have revised the Introduction to clarify that our argument concerns the relative distinctiveness of the climate context:

“Although many social movements involve multiple and intersecting fronts, debates around climate change are arguably marked by particularly diffuse and shifting intergroup boundaries. Rather than coalescing around a single, historically entrenched identity cleavage, climate action draws upon — and cuts across — political, national, generational, and sectoral fault lines. This fluidity means that potential participants are not always sure who they are standing with or against, which may dampen the potency of traditional group-based drivers of action (e.g., strong identification with an aggrieved ingroup versus antagonism toward a clear outgroup). Thus, it remains an open and important question whether motivational pathways established in more identity-centric movements apply fully to the climate context.”

Third, some claims in the introduction and conclusions in the discussion appear decontextualised. The fact that the data were collected from Australian participants is not properly mentioned until the method section. This omission gives the impression that the geographic and sociopolitical context has no bearing on the study’s conclusions or arguments. Greater acknowledgement of how the Australian context might shape the findings is crucial.

We agree that sociopolitical context can significantly shape predictors of collective action, and acknowledge that we did not sufficiently foreground the Australian setting early in the manuscript. We have revised the Introduction to more explicitly locate the study within Australia’s political and cultural climate change context and to acknowledge that this context may influence the generalisability of our findings.

“This study was conducted in Australia — a context that has historically experienced high levels of public concern about climate change alongside protracted political polarisation and

policy volatility. Such dynamics may shape both the salience of climate action and the psychological drivers of activism intentions, and we return to these contextual factors in interpreting the findings.”

We also added a note in the Discussion reflecting that replication across additional geopolitical contexts would strengthen confidence in the broader conclusions.

“It is also important to interpret our findings in light of the Australian context in which the study was conducted. Australia represents a sociopolitical environment characterised by high public awareness of climate change but historically polarised political leadership and policy inconsistency. These contextual dynamics may influence both the strength and nature of the psychological processes examined here, and future research would benefit from testing whether the patterns observed replicate across different geopolitical and cultural settings.”

Fourth, the authors present several distinct theoretical perspectives under the broad umbrella of SIMCA (Social Identity Model of Collective Action) and its extensions. However, this approach risks being reductive, particularly in the case of DIME and MOBILISE. These models do not merely extend or elaborate on SIMCA; they also explore the dynamic processes that sustain mobilization. While they are influenced by SIMCA, they incorporate broader mechanisms beyond psychological predictors, which should be acknowledged.

We agree that while DIME and MOBILISE are influenced by SIMCA, they go beyond SIMCA by conceptualising mobilisation as a dynamic, temporally unfolding process, and by incorporating a broader set of mechanisms (e.g., tactical innovation, resource flows, feedback loops). We have revised the Introduction to clarify that these models are not simply extensions of SIMCA but draw upon SIMCA-derived insights while emphasising additional processes that sustain mobilisation over time.

“Recent theoretical developments — for example the disidentification, innovation, moralization and energization model (DIME)¹⁵ and the model of belonging, individual differences, life experience and interaction sustaining engagement (MOBILISE)¹⁶ —build on insights from SIMCA while also emphasising dynamic and processual aspects of mobilisation such as tactical innovation, resourcing, and feedback loops.”

Finally, throughout the manuscript, several claims require additional citation support. Some statements appear speculative or broad, and reinforcing them with relevant literature would strengthen the argumentation.

We agree that in several places the argumentation would benefit from stronger empirical grounding. We therefore reviewed the manuscript carefully to identify any statements that were overly speculative or broad, and added appropriate supporting citations to reinforce key points in the Introduction and Discussion.

Reviewer #2

The present article “Rethinking the psychology of support for radical climate activism” presents data from a cross-sectional sample of the Australian public that supports climate action in an attempt to test how traditional theories of collective action (e.g., SIMCA) align with normative and non-normative ‘radical’ action. Specifically, the article attempts to study this in the domain of climate change, where ingroup and outgroup identification becomes

difficult to disentangle, resulting in questions about the validity of these theoretical constructs in this domain. While I think the questions raised in this piece are worthwhile areas of study, I do have concerns about the sample, conceptualization, and analysis. Many of these can be addressed through edits to the current manuscript, including additional details/analyses. However, to answer the questions posed in this manuscript, I think additional data may be required.

We thank Reviewer 2 for the supportive comments and the thoughtful suggestions for improvement.

More information needed about the data

I appreciate the descriptive statistics that the authors present throughout the manuscript, that characterize things like overall support for normative and radical collective action. Additionally, the authors present the alpha and correlation coefficients for all scales in supplement, which is helpful to the reader to understand these measures. However, I would request that they also report the mean and SD so readers can evaluate these measures more. Particularly as some may have more limited or skewed distribution and possible ceiling effects.

We have now included overall means and standard deviations in Table 2.

Also, there is a lack of data visualization that seems key in a cross-sectional analysis like this. We're shown group differences and regression coefficients. But the distribution of variables (particularly the support variables) is important to see. This may be particularly important for some associations, such as belief in climate change and support for radical climate action. Showing histograms, scatterplots, violin plots, etc. can help illustrate the nature of the relationship, and allow for critical analysis of if there are issues in the data (e.g., extreme outliers).

In the supplementary file we now include 21 cloud plots visualising the distribution and central tendencies of the 19 predictors and two outcome variables at Wave 1. Doing so means that readers can inspect the distributions directly and assess potential skew or ceiling/floor effects. With 19 predictors across two models, it is not feasible to visualise every bivariate association in the main text; many of which would not map directly onto the multivariate analyses reported. Instead, we have supplemented the regression analyses with independent-samples t-tests comparing supporters versus non-supporters of radical action. These t-tests converge with the regression results and are less vulnerable to undue influence from influential individual data points, providing additional reassurance about the robustness of the findings.

One final note about additional information needed from the data, is that there should be some additional information about support for normative and radical action. These are the key outcomes of this study, and are treated only as composite measures. While both have very high internal reliability, I think two types of information would be helpful in evaluating them. First, factor analysis to show a) that these are two distinct composite measures and b) that there aren't subfactors or groupings of types of action within each. Second, particularly for the normative collective action, people may support some actions more than others. Descriptive information for each item could be useful for the reader and add another layer of detail to your arguments (though this does not need to be in the main

text).

We have included a new section in the supplementary file that reported factor analyses confirming two factors corresponding to conventional and radical collective action (Table S3). We have also added a new Table S4 which summarises means and standard deviations for each item of the collective action scales separately.

Concerns about measurement and conceptualization

The authors made a good effort to include many relevant factors in their analysis. However, I think there are some issues with their conceptualization and/or measurement tools used. These issues may not invalidate their findings but gives several caveats to the conclusions they can draw.

First, the primary outcome measures are consistently referred to as ‘support’. However, these are behavioral intentions. This is an issue for two reasons. First, intentions require intentions to act, which relies on many other factors (e.g., people may feel that they don’t know how or where to start, what to do, that they cannot physically attend protests or engage in violent protest). So, the predictive factors here that should lead to action may be mitigated by external factors, leading the intention-behavior gap to be quite large. For example, there was a good amount of ‘support’ for normative collective actions. However, only a minority of people regularly (if ever) engage in actual protests. This could mean that, while the factors in the model predict intentions, they may not actually predict behavior. Second, support is actually an important part of protest, in that bystanders and the general population’s approval of collective action is an important part of creating change. As such, I would urge the authors not to conflate support and intentions.

Our use of the term support was intended in the colloquial sense of supporting climate-related collective action; however, as the reviewer rightly observes, the operational items in this study explicitly refer to respondents’ own intentions to engage in collective action (e.g., “I intend to join...”, “I intend to volunteer...”, “I intend to participate...”). To avoid confusion, we have revised the manuscript to refer consistently to “(collective) action intentions” rather than “support”. We have also added a note in the Discussion acknowledging that intentions do not always translate into behaviour due to behavioural barriers, and that future work is needed to examine the extent to which the predictors identified here forecast actual participation.

Secondly, identification is a key part of many previous models (e.g., SIMCA, SIMPEA), so its inclusion is well warranted. However, there are a few issues with how it is measured. First, the measurement tool is mostly reverse scored items, so is it a measure of identification or a measure of social distance from opponents of the movement? It also is not identification with a specific movement but with ‘other people who have similar views on climate change as you do’ (whatever those views are). This means that it may be either a measure of identification with the broad swath of the population that supports climate action in principle (an overly broad identification measure that may not be particularly motivating) or disidentification with the opposition. Given this, it’s not unsurprising that it does not show significant associations with radical action. Particularly as radical or progressive actors may already distance themselves from the rest of the general pro-climate public (see the vegan vs. vegetarian dynamic), but instead identify with other, similar, radical climate actors. I agree that an ingroup-outgroup dynamic is difficult to concisely conceptualize in this domain, but it

is hard to say that ingroup matters or doesn't when it's such a broad definition.

Our revised analyses incorporate Wave 2, which measured identification using a single (positively worded) item. This means that we are no longer relying so heavily on negatively worded items (i.e., it is more of a traditional identification scale rather than a scale that incorporates elements of disidentification). Perhaps because of this, the new analyses now show a positive relationship between identification and radical collective action intentions. The manuscript has been revised throughout to adjust for these new conclusions.

I also would appreciate more discussion on efficacy specifically. There are many forms of efficacy (self, collective, contributive). The form of efficacy measured here does not relate to radical action intentions. However, that isn't surprising if radical actions are seen as an individual action meant to raise awareness or cause harm rather than solve the problem directly.

We thank the reviewer for this insightful point. Our efficacy measure focused specifically on collective efficacy: respondents' beliefs that people concerned about climate change, acting together, can influence outcomes. We acknowledge that this represents only one form of efficacy and differs from contributive efficacy (belief that one's own participation will make a meaningful difference), which may be particularly relevant for predicting willingness to engage in radical or disruptive forms of action aimed at raising awareness rather than solving the problem directly. We clarify this distinction in the manuscript and note, in the Discussion, that future research is needed to examine how different forms of efficacy may differentially predict normative versus radical action intentions.

“Fourth, despite theorising linking radical action to perceptions of efficacy, no reliable association was found in our data. Notably, our measure captured collective efficacy (beliefs about what people concerned about climate change can achieve together), rather than contributive or self-efficacy (beliefs about the impact of one's own involvement). It is possible that these other forms of efficacy may be more strongly implicated in willingness to engage in disruptive or radical forms of action, a question that future studies could address directly.”

Analyses

I appreciate the use of alternative analysis methods. However, the paper feels fairly brief as is, and the re-analysis (multilevel model vs. regression) does not seem to add new content. I might suggest that the authors consider presenting only one of those types of models in the main text, and reporting the rest in supplement (however this is not a critical issue).

I would however suggest an additional analysis for the authorial team to present. The multilevel model currently treats support at the different time points simply as indicators to an underlying latent trait of 'willingness to take action' across time. However, this dataset gives the authors a chance to actually analyze these results longitudinally, making their contribution much more significant. If there is change in support across the timepoints of the study, treating that at a linear model and using the various measures to predict changes in support would seem to be a greater indicator of actual intention/behavior shift. Alternatively, if that model does not converge, using difference scores from time 1 to time 2.

We thank the reviewer for this constructive suggestion. In our original submission we only reported on two waves of data, which limited our ability to examine longitudinal change directly. We now include a third wave, which enabled us to run Dynamic Structural Equation Models (DSEMs). DSEM extends the random-intercept cross-lagged panel model by distinguishing stable, trait-like differences between individuals from within-person changes across time. In practice, this allowed us to test whether fluctuations in key psychological predictors forecast later changes in support for collective action.

Because some predictors were measured only at baseline (e.g., personality, political ideology), and because models with all longitudinal predictors included together did not converge, we focused the DSEMs on the subset of predictors that were significant in our multilevel models. We estimated separate bivariate DSEMs for these variables, restricted to participants who completed all three waves. In sum, the addition of these DSEM analyses strengthens the longitudinal contribution of the manuscript, consistent with the reviewer's suggestion to examine change over time rather than treating support as a static trait.

Given the possibilities offered by the longitudinal dataset, we agree with Reviewer 2 that the regressions originally reported for the Wave 1 analyses are adding little value and so have deleted these from the revised manuscript.

Regardless of whether it is analyzed in a longitudinal fashion or not, I do not think it is acceptable to not present the intermediate survey results. Assuming there is no change in the pattern of findings, then there's no reason to omit them. And if there is a change, then it becomes especially critical to show those effects. Omitting reporting this time point presents concerns about cherry-picking ideal contrasts across specific times points. If the intermediate set of data didn't contain all variables, you should use only those present or use the predictors from time 1. Also, the argument that there are high levels of noise is unclear. How was this assessed? I suggest presenting these results, and if you want to make an argument about statistical noise, present the data that shows that they should be omitted. This can all be presented in supplement, but it shouldn't be omitted.

We initially omitted Wave 2 for two reasons: (1) high (and hard-to-explain) levels of statistical noise in Wave 2 and (2) the fact that only abbreviated versions of the predictor scales were administered at Wave 2, which complicated direct comparability across all three time-points. The first issue is no longer a problem. In the process of preparing this revision, we discovered an administrative error affecting the data matching between Waves 1 and 2 involving nearly half the Wave 2 respondents. This problem has now been resolved. (As an aside, in resolving this problem we corrected an error that was present in the original submission. Specifically, to enforce our policy of only analysing participants 18 years old or over we had inappropriately removed from analysis 80 participants who had missing data on age. These participants are now included in the analyses in the revised manuscript).

Resolving the second issue required us to use the shorter scales that were administered at Wave 2 for all the waves. This carries a cost psychometrically, but we agree that including the Wave 2 results enhances transparency and addresses potential concerns around selective reporting. Importantly, the pattern of findings in the new analyses is broadly compatible with the old analyses (that just used the two waves), strengthening confidence in the robustness of the results. There are two exceptions. First, incorporating Wave 2 required us to drop positive emotions from our analysis because Wave 2 used entirely different positive emotion item to those used in Waves 1 and 3. Second, previously non-significant effects of moral conviction,

identification, and anger on radical collective action intentions are now (weakly) significant. This is more aligned with theoretical expectations, and has required us to adapt one of the messages of the original paper, which is that SIMCA variables mostly don't predict radical action intentions. Note, however, this conclusion is still true of collective efficacy, and (as noted above) the longitudinal analyses show little support for the traditional SIMCA variables in predicting radical collective action intentions.

On a larger point though, even with these alternative analyses, the issue is still cross-sectional, examining intentions in a broad section of the Australian public that may never actually follow-up on these intentions. As such, this one study is hard to draw conclusions from. My suggestion to the authors would be to add to the study, adding further evidence that supports their findings of what motivates radical and normative climate action (e.g., experimental data that manipulates the theorized constructs that motivate radical action, cross-sectional analysis on actual climate activists, qualitative data, etc.)

Now that we have included the third wave of data, the study is no longer cross-sectional. Although extra data can always add empirical weight to a story, we believe our study already offers a substantial contribution, tracking a sample of more than 1,400 climate change supporters across three time-points over 12 months. Importantly, the editor did not indicate that additional data collection was required for revision, and the original grant under which the study was conducted has now concluded. We have therefore focused our revision on enhanced transparency and robustness in reporting (e.g., the full Wave 2 analyses) and adding longitudinal analyses rather than expanding the design. In the Discussion we have now added the following: "Although the longitudinal design employed here strengthens inference relative to single-wave cross-sectional research, future studies using experimental manipulations or activist samples would be valuable in triangulating and extending the pathways identified in the present work."

Miscellaneous points

In the abstract, the authors say that "These findings challenge media portrayals of radical climate activists". What is meant by this?

We have now removed that phrase from the abstract.

The authors say that there is moderate support, but only 25% of the sample were above the midpoint. This seems like a poor way to describe as opposed to moderate or broad opposition.

We have now altered this text to read "Overall conventional collective action intentions were relatively limited ($M=3.16$, $SD=1.28$), but there was substantial variability in views, with approximately a quarter of respondents (25.3%) scoring above the scale midpoint."

There was no measure of prior political or activist engagement? That seems like a critical variable to omit given the past literature on the important role of attending events and engaging with activists in actually mobilizing action.

We agree with the reviewer that prior political or activist engagement is an important predictor of participation in collective action. Unfortunately, we did not include a direct measure of this in our survey, which is a limitation of the present study. Our focus was on

psychological and demographic predictors of collective action intentions in a community sample, rather than on activist sub-populations, but we acknowledge that prior engagement could further shape attitudes and behaviours. We have now noted this as a limitation in the manuscript and flagged it as a valuable direction for future research: “Another limitation of the present study is that we did not measure participants’ activist engagement, which the literature suggests may be an important factor shaping support and participation. Future research should incorporate this dimension to provide a fuller picture of mobilisation processes.”

“Interestingly, perceptions that other supporters felt hostility toward opponents reliably predicted radical collective action support, even when respondents’ own feelings of hostility did not.” But you don’t measure hostility. If you’re referring to things like empathy, that’s a different measure.

In the 3-wave analyses the effect of norms of hostility was non-significant, so this sentence has been deleted.

Please report average political leaning after exclusions (rows 431 to 434)

We now report average political leaning after exclusions.

Only some of the data and script are shown in the OSF link. None of the MPlus script is there.

The OSF link now contains all the relevant code including the MPlus script.

For all supplemental tables, it would be helpful to bold the significant effects, as you did for Table S7.

Significant effects are now bolded.

One point that might be missed in radical versus collective action is the call for retributive vs. restorative justice, in regards to outgroups. If the outgroup is someone who caused the issue or proved to be an obstacle to solving it, should beliefs about punishment as justice predict willingness to take violent (versus peaceful) action?

As you’re looking at conflict literature and applying it to this area of work, I would suggest ‘conflict narratives’ as a potential future direction or other factor relevant in this work. As the climate movement has been ongoing for decades, people may have developed narratives about the nature of the conflict that can give them meaning in the conflict and fuel radicalization. (e.g., Ulug et al., 2021, How do conflict narratives shape conflict- and peace-related outcomes among majority group members).

We appreciate the reviewer’s suggestion to consider the role of retributive versus restorative justice orientations, and the broader notion of conflict narratives, in shaping support for radical versus conventional action. While these variables were outside the scope of our current dataset, we agree they represent a promising direction for future research. In particular, they may intersect with the dynamics of collective victimhood highlighted in our study: when people perceive their ingroup as victimised, they may be more inclined toward retributive frames that justify radical tactics, whereas restorative frames may be more

consistent with conventional, cooperative forms of action. We have added a sentence in the Discussion noting this as a potential avenue for future work: “Future research could also examine how justice orientations and conflict narratives shape responses to climate action. For instance, perceptions of collective victimhood may be linked with retributive frames that justify punitive or radical tactics, whereas restorative frames may channel support toward more conventional, cooperative action (Ulug et al., 2021).”

In total, I like this paper and the arguments it is trying to make. The difference between normative and non-normative (radical) action is a topic that, while previously explored, is still relatively omitted from psychological literature. Expanding existing theories to address this distinction is key, particularly in the domain of climate action where intergroup lines and dimensions can become nebulous. This is a critical area of exploration for our field. However, I have reservations about the manuscript in its current form. To further support the conclusions of this study, I would suggest the inclusion of additional data. However even as is, I think there are some edits that can be made to strengthen the paper, address concerns about the data, and modify the conclusions that can be drawn from the analysis.

We are grateful for the reviewer’s thoughtful engagement with our work and for highlighting both the promise and the challenges of this research area. We share the view that distinguishing between normative and non-normative forms of climate action is a critical step for advancing collective action theory, and we are pleased that the reviewer sees value in our contribution. In revising the manuscript, we have sought to address the reviewer’s concerns in several ways: (1) we added a third wave of data and conducted Dynamic Structural Equation Models to directly test longitudinal processes, as recommended; (2) we clarified the distinction between intentions and participation, and our rationale for focusing on intentions; (3) we acknowledged limitations such as the absence of measures of prior activist engagement; and (4) we incorporated suggestions for future research directions, including the potential role of conflict narratives and justice orientations. We hope these revisions and clarifications substantially resolve the reviewer’s reservations and strengthen the contribution of the paper.

RESPONSE TO REVIEWERS

Reviewer #1

There are still some substantial issues that remain unaddressed. First, the measurement of collective action combines multiple constructs without sufficient theoretical justification. Some items measure support for collective action, while others assess collective action intentions. The provided factor analysis, in its current form, does not sufficiently warrant this combination. Also, two cross-loading items were not discussed or reflected upon in either the methodology or discussion, which raises concerns about construct validity. There is a clear mismatch between how these constructs are introduced in the introduction and how they are operationalised in the study. For example, some items labelled as “radical collective action” (such as chaining oneself to a fence) are not necessarily law-breaking or violent, depending on the context.

We thank the reviewer for this comment. We respond to each element in turn and have clarified our measurement decisions in the manuscript and supplementary materials accordingly.

Conceptual distinction between “support” and “intentions”: Our study deliberately focused on intentions to engage in various forms of collective action, consistent with the large majority of psychological research using intentions as proxy indicators of collective action involvement (see Agostini & van Zomeren, 2021, meta-analysis on this point). The full wording of items (highlighted in Supplementary Table S1) reflects this: every item explicitly begins with “I intend to...” whether referring to joining, donating to, or participating in conventional or radical collective action. As can be seen in Table S1, there are no items measuring attitudinal support; all items explicitly address the intention component.

Empirical justification from the factor analysis: Our measurement approach was adapted based on the approach developed by Moskalenko & McCauley (2009). As reported in Supplementary Table S3, the principal components analysis with oblimin rotation yielded a clean two-factor solution, with all conventional items loading strongly on the first factor and all radical items loading strongly on the second factor. These two factors jointly explained 68.5% of variance. Across the 13 items, only two showed meaningful cross-loadings. We now explicitly discuss these cross-loadings in the Methods section, including a discussion of why these two items behaved in ways that are theoretically interpretable:

“As shown in Supplementary Table S3, the exploratory factor analysis produced a clear two-factor solution corresponding to conventional and radical collective action intentions. All items loaded most strongly on their theorised factor, with two items showing secondary cross-loadings of interpretable magnitude. The “participate in a sit-in” item loaded primarily on the conventional factor ($\lambda = .58$) but also showed a modest secondary loading on the radical factor ($\lambda = .40$), consistent with the ambiguous status of sit-ins in the literature, where they are variously classified as conventional, disruptive, or borderline non-normative depending on context. Likewise, the item “donate to an organisation that sometimes breaks the law” loaded primarily on the radical factor ($\lambda = .51$) but showed a weaker secondary loading on the conventional factor ($\lambda = .43$), reflecting its combination of a conventional action form (donation) with radical organisational behaviour. Because both items loaded most strongly on their intended factor, contributed to high internal reliability within each

scale, and mapped onto theoretically coherent distinctions, they were retained in accordance with recommendations for construct-valid scales with conceptually interpretable cross-loadings.”

Clarifying the classification of “chaining oneself to a fence”: Our intention was to capture a set of behaviours widely coded in the literature as non-normative, disruptive, or law-breaking. We note that our analytic distinction is based on normativity and legality, not solely on violence. The item “chain oneself to a fence in protest” loaded cleanly on the radical factor, consistent with its classification as a clearly rule-breaking behaviour. This is particularly the case in the climate domain, where such tactics are typically used to halt fossil-fuel infrastructure activities and are widely considered illegal and non-normative.

Second, the literature review remains rather superficial. In recent years, numerous studies and theoretical discussions have advanced our understanding of radical collective action and resistance. Focusing solely on SIMCA and other mainstream approaches provides only a very general overview. Given the current depth and sophistication of the literature, such a broad and limited review does not adequately situate the study within contemporary theoretical debates. Furthermore, the predictors included in the study are not theoretically introduced or justified. Without a grounded discussion of how and why certain variables might differentially relate to conventional versus radical actions, the theoretical framing remains underdeveloped, and the inclusion of these variables appears somewhat arbitrary. There are lots of new insights and debates in the literature on the role of identity and efficacy (particularly, what kind of identification and what kind of efficacy actually lead to radical over conventional collective action), which authors do not engage in any of these discussions.

We thank the reviewer for this thoughtful and generative comment. We understand the perception that the initial literature review placed too much emphasis on SIMCA and did not sufficiently engage with contemporary debates surrounding radical versus conventional collective action. In response to this perception, we have broadened and deepened the theoretical framing in the Introduction. For example, we have now added this text as the new paragraph 2:

“A long tradition of research has distinguished between forms of collective action that are *conventional*—that is, lawful, socially sanctioned, and aligned with prevailing norms—and *radical* actions that are illegal, disruptive, violent, or otherwise outside institutional rules (Wright et al., 1990; Louis et al., 2020). This distinction closely parallels terminologies reflecting differences between *normative versus non-normative* (Becker & Tausch, 2015) *activism versus radicalism* (Moskalenko & McCauley, 2009), *legal versus illegal* (Finkel et al., 1989), and *hostile versus benevolent* action (Zaal et al., 2011). Importantly, prior work shows that different forms of action are shaped by different psychological pathways. For example, Selvanathan and Leidner (2020) demonstrate that normative and non-normative actions emerge from distinct constellations of identity and justice concerns: ingroup attachment predicts normative action via restorative justice motives, whereas ingroup glorification predicts non-normative action via retributive motives. The distinction has also been explored in the environmental domain: Bashir et al. (2013) find that the public differentiates between “typical” and “atypical” environmentalists, and that atypical (i.e., more extreme or radical) actors elicit social distancing and reduce willingness to engage in or support environmental action. Although various terminologies have been used across domains, they converge on a shared insight:

people appear to draw categorical distinctions between conventional actions that work *within* established systems and radical actions that seek to disrupt or challenge them. What remains unclear, particularly in the climate context, is whether these distinctions reflect incrementally stronger engagement along a single continuum, or whether they represent qualitatively distinct forms of action associated with different motives. Our study directly addresses this gap.”

We have also expanded the Discussion to engage directly with the reviewer’s point that contemporary literature offers more nuanced accounts of efficacy and identification than were reflected in the original submission. We now explicitly acknowledge that our measures captured broad or general forms of these constructs, and we discuss how emerging theoretical distinctions—such as collective versus contributive efficacy, and politicised, moralised, or fused forms of identification—may yield more fine-grained insights into radical versus conventional forms of climate action. The new paragraphs emphasise that future research should examine whether these differentiated constructs better predict willingness to engage in disruptive or high-cost action. In doing so, we connect our findings with the “depth and sophistication” of current debates, without overstating what our data can resolve.

“Fourth, despite theorising linking radical action intentions to perceptions of efficacy, no reliable association was found in our data. This contrasts with the findings for conventional collective action intentions, where efficacy was the only variable to predict both cross-sectionally and longitudinally. Notably, our measure captured *collective efficacy* (beliefs about what people concerned about climate change can achieve together) rather than contributive or self-efficacy (beliefs about the impact of one’s own involvement). It is possible that different forms of efficacy may be more strongly implicated in willingness to engage in disruptive or radical forms of action, a question that future studies could address directly.

A similar point applies to social identification. In the present study we used a widely validated single-item measure capturing general identification with others who share one’s climate change views. However, recent theoretical developments distinguish between different forms of identification—such as politicised identity, moralised identity, and identity fusion—which may have distinct implications for normative versus non-normative collective action (Jasko et al., 2017; Swann et al., 2014). Whereas broad identification typically predicts conventional engagement, more intense or fused forms of attachment have been linked in some studies to willingness to engage in high-risk or costly collective behaviour. Future research could therefore test whether specific types of efficacy and identification, rather than general measures of these constructs, differentially predict the more disruptive or confrontational behaviours that constitute radical action. By incorporating these constructs, our study aligns with contemporary debates emphasising that different types of identification, efficacy, and grievance-based emotions may give rise to distinct forms of collective action rather than a single escalation continuum.”

Finally, to avoid the impression that predictors were included in an ad hoc or atheoretical manner, we note that Table 1 shows how each variable maps onto multiple theoretical traditions: not only SIMCA but also research on identity, victimhood, moral conviction, emotions, and perceptions of state power. This makes clear that the study is not anchored in a single model but draws on a broader constellation of theoretical perspectives that have been applied in the study of both normative and non-normative collective action.

Taken together, these revisions provide a more comprehensive theoretical framework, situate the study more clearly within contemporary debates on radical action and resistance, and explicitly acknowledge the conceptual distinctions that future work should explore in more detail. We are grateful to the reviewer for prompting these improvements, which we believe have strengthened the manuscript considerably.

Reviewer #2

The revisions have largely addressed my previous concerns, and I think the current version of the manuscript could be suitable for publication. However, there are a few issues that still need to be addressed.

First, the provision of the cloud plots for the outcome measures is helpful, but it does further highlight the intensive skew and floor effect on radical action intentions. While that skew is noted in the manuscript, I think there are additional limitations to be noted. First, we can see that a supermajority of responses are below a 2 on the scale. While it's true that an exceedingly small minority of the public will ever engage in radical, violent protest, this accentuates the issue of the gap between action and intentions. Intentions may be low across the board but noting that other social factors such as deindividuation and mob mentality may be more important in promoting action than reasoned intentions. This skewness can harm what inferences we can make about linear associations between this factor and other variables. And the t-tests try to accommodate for this, but the comparison group (everyone above the midpoint) includes people who still outright disagree with most of these items (i.e., a 2-3 on the scale). As such, it may be worth considering a different cutoff point in relation to the scale's values (average of 3+ for example) to compare outright disapproval to approval. However, this further limits the sample size for these comparisons.

Given that this is the primary takeaway from this paper (i.e., what does or does not predict radical action intention) this is a major issue. Without meaningful variability in this outcome, it is difficult to draw conclusions about predictors/differences. The DSEM model may alleviate this, however, it found no significant predictors at all.

As such, this limitation needs to be noted. The authors currently discuss in the discussion, without hedging their claims, the various associations with radical action intentions, and do not mention this issue of skew or limited variance in their limitation section of the discussion. The only place it is mentioned is as a caveat to their non-significant DSEM findings. As written, this frames that lack of findings as potentially untrustworthy, but their other findings as relatively certain. Discussion of what was found to predict conventional collection action intentions is suitable and warranted, but the other findings should be severely limited.

We thank the reviewer for this comment. We respond to each element in turn.

Clarifying a misunderstanding: We appreciate the concern that our t-test comparison group may include respondents who are still very low in radical action intentions. However, we would like to clarify that we did not employ a median split nor compare “low vs. high” based on sample distributions. Instead, respondents above the true midpoint of the response scale (i.e., > 4 on the 1–7 scale) were classified as the “higher-intention” group.

The reviewer writes as though we set the cutoff at the sample median, which would indeed sweep in respondents scoring 1 or 2. But the midpoint cutoff only captured respondents who were consistently scoring well above population norms on these items. These respondents

averaged strong agreement on some radical items (e.g., vandalism, blockades) and thus represent genuine endorsement of radical tactics.

Floor effects and skewness in radical action intentions: We agree with the reviewer that radical climate action intentions are skewed and near-floor in the general population. This is theoretically expected and is also consistent with the exceptionally low base rates of radical action documented in prior work – and in society at large! However, we accept that the consequences of this skew should be discussed more explicitly.

For example, we agree that the DSEM null findings should not be framed as evidence against within-person effects but may in fact reflect limited variance over time, floor effects, and the theoretical reality that radical intentions among supporters of climate action are fairly stable, low-frequency orientations. In line with this, we have now included the following text in the final paragraph of the Results section:

“Interestingly, none of the longitudinal analyses involving radical collective action intentions were significant (see Supplementary Table S6). As shown in our cloud plots (Fig. S3) and descriptive statistics (Table S4), radical intentions displayed an extreme floor effect, with the majority of respondents scoring below “2” on a 1–7 scale and only a very small proportion endorsing these behaviours. This restricted variance limits the inferences that can be drawn about predictors of radical intentions, and it likely contributes to the absence of longitudinal effects in the DSEM models. Radical climate action is a genuinely low-base-rate phenomenon, and rare behaviours measured via self-reported intentions are particularly susceptible to distributional constraints. As such, the DSEM null findings should not be framed as evidence against within-person effects but may in fact reflect limited variance over time, floor effects, and the theoretical reality that radical intentions among supporters of climate action are fairly stable, low-frequency orientations.”

In the opening paragraph of the Limitations section of the Discussion, we have also added the following:

“Although the multilevel models identified several variables associated with higher radical action intentions, these findings must be interpreted cautiously. Radical intentions were extremely low and highly skewed across the sample, producing restricted variance that necessarily constrains the precision of linear estimates. The significant predictors we observed—particularly collective victimhood and lower conscientiousness—showed convergence across analytic approaches, but these associations should be understood as preliminary rather than definitive.”

In line with Rev 1’s suggestions, we have also added the following text to the paragraph in the Discussion talking about the limitations of intentions:

“Moreover, intentions may not fully capture the situational and group-level processes (e.g., deindividuation, crowd dynamics, heightened arousal) that sometimes are associated with radical behaviour. These processes are unlikely to be reflected in stable individual differences measured via a panel survey, and may account for additional variance in real-world radical action beyond what can be modelled here.”

Second, there is need for additional context in the academic literature in regards to the differences between radical and conventional collective action intentions. The authors mention some other terminology that has been used to describe differences in these types of action (Introduction paragraph 2, lines 37-42) but do not cite past work. For example, they mention ‘normative’ and ‘non-normative’ collective action without contextualizing how their concepts are the same or different from this and/or what the findings of those past studies are. For example, Selvanathan & Leidner, 2020) found different effects of attachment and glorification on normative and non-normative collective action support, mediated by different paths of retributive versus restorative justice. As another example, members of the lay public had different perceptions of ‘typical’ and ‘atypical’ environmentalists, resulting in differences in intentions and support through desired social distance (Bashir et al., 2013). While neither of these papers put forth a specific model (e.g., SIMCA), both clearly have connections to this distinction. Further explanation of what matters in this difference (e.g., legality, normativity, violence, disruption, prototypicality) and how that has been explored in the climate domain and other domains is necessary.

We thank Reviewer 2 for this thoughtful comment and for directing us to those papers. We have now added the following paragraph which acknowledges that literature and contextualises our discussion of radical versus conventional action within a broader literature.

“A long tradition of research has distinguished between forms of collective action that are normative—that is, lawful, socially sanctioned, and aligned with prevailing norms—and nonnormative actions that are illegal, disruptive, violent, or otherwise outside institutional rules (Wright et al., 1990; Becker & Tausch, 2015). This distinction closely parallels terminologies used in neighbouring literatures, such as *conventional versus radical* (Moskalenko & McCauley, 2009), *legal versus illegal* (Finkel et al., 1989), and *hostile versus benevolent* action (Zaal et al., 2011). Importantly, prior work shows that different forms of action are shaped by different psychological pathways. For example, Selvanathan and Leidner (2020) demonstrate that normative and non-normative actions emerge from distinct constellations of identity and justice concerns: ingroup attachment predicts normative action via restorative justice motives, whereas ingroup glorification predicts non-normative action via retributive motives. The distinction has also been explored in the environmental domain: Bashir et al. (2013) find that the public differentiates between “typical” and “atypical” environmentalists, and that atypical (i.e., more extreme or radical) actors elicit social distancing and reduce willingness to engage in or support environmental action. Although various terminologies have been used across domains, they converge on a shared insight: people appear to draw categorical distinctions between actions that work *within* established systems and those that seek to disrupt or challenge them. What remains unclear, particularly in the climate context, is whether these distinctions reflect incrementally stronger engagement along a single continuum, or whether they represent qualitatively different forms of action underpinned by different psychological profiles. Our study directly addresses this gap.”

I still think there are some interesting findings to be presented from this paper. For example, even the descriptive findings about levels of intentions for conventional and radical actions is an interesting statistic worth reporting. Similarly, the factor analysis could highlight meaningful differences in understandings/intentions of different types of actions that can further supplement past research and help in situating their findings in past literature that may not have done empirical measurement work to distinguish these types of acts. However,

without additional discussion of past findings in the literature we are left without clear guidance on how this fits with other (currently uncited) works. And heavy edits are needed to correctly hedge all claims regarding predictors of radical action intentions given the critical measurement issues.

We thank the reviewer for these generous and helpful observations. We agree that the descriptive findings—including the marked contrast between levels of conventional and radical action intentions—represent an important contribution in their own right. We have also strengthened the Discussion by incorporating additional relevant literature and clarifying how our results complement or diverge from prior studies. Finally, as noted above, we have substantially revised the framing of claims regarding predictors of radical action intentions to ensure they are appropriately hedged given the distributional constraints of the measure.

RESPONSE TO REVIEWERS

Reviewer #1

I believe my concerns are sufficiently addressed. I only want to raise one minor issue. Please support your claims with citations for the following argument:

The “participate in a sit-in” item loaded primarily on the conventional factor ($\lambda = .58$) but also showed a modest secondary loading on the radical factor ($\lambda = .40$), consistent with the ambiguous status of sit-ins in the literature, where they are variously classified as conventional, disruptive, or borderline non-normative depending on context.

We have now added three references to support this point.

Shuman, E., Saguy, T., van Zomeren, M., & Halperin, E. (2021). Disrupting the system constructively: Testing the effectiveness of nonnormative nonviolent collective action. *Journal of Personality and Social Psychology*, 121(4), 819–841.
<https://doi.org/10.1037/pspi0000333>

Sharp, G. (2020). *The politics of nonviolent action*. The Albert Einstein Institution.

Beer, M. A. (2021). *Civil resistance tactics in the 21st century*. International Center on Nonviolent Conflict. <https://www.nonviolent-conflict.org/wp-content/uploads/2021/03/Civil-Resistance-Tactics-in-the-21st-Century-Monograph.pdf>

The first is the most relevant, since it describes sit-ins as a nonnormative nonviolent tactic, placing them in that kind of in-between space between conventional and radical. The other two are sources that more broadly speak about sit-ins, as part of the activist tactical repertoire. Gene Sharp described sit-ins as a tactic of “nonviolent intervention”, and the book describes many examples of sit-ins that help backs up what we have said about them being able to be categorised differently depending on context. The final source is simply considered an update to Sharp’s “198 methods of nonviolent action” outlined in his book, situating sit-ins amongst many other tactics that activists employ.